# Neural Fractional Attention Differential Equations

**Qiyu Kang**[1][*], **Wenjun Cui**[2, 5][*], **Xuhao Li**[3][†], **Yuxin Ma**[1],
**Xueyang Fu**[1], **Wee Peng Tay**[4], **Yidong Li**[5], **Zhengjun Zha**[1]
[1]University of Science and Technology of China [2]Shanxi University [3]Anhui University
[4]Nanyang Technological University [5]Beijing Jiaotong University

## Abstract

The integration of differential equations with neural networks has created powerful tools for modeling complex dynamics effectively across diverse machine learning applications. While standard integer-order neural ordinary differential equations (ODEs) have shown considerable success, they are limited in their capacity to model systems with memory effects and historical dependencies. Fractional calculus offers a mathematical framework capable of addressing this limitation, yet most current fractional neural networks use static memory weightings that cannot adapt to input-specific contextual requirements. This paper proposes a generalized neural Fractional Attention Differential Equation (FADE), which combines the memory-retention capabilities of fractional calculus with contextual learnable attention mechanisms. Our approach replaces fixed kernel functions in fractional operators with neural attention kernels that adaptively weight historical states based on their contextual relevance to current predictions. This allows our framework to selectively emphasize important temporal dependencies while filtering less relevant historical information. Our theoretical analysis establishes solution boundedness, problem well-posedness, and numerical equation solver convergence properties of the proposed model. Furthermore, through extensive evaluation on tasks such as fluid flow, graph learning problems and spatio-temporal traffic flow forecasting, we demonstrate that our adaptive attention-based fractional framework outperforms both integer-order neural ODE models and existing fractional approaches. The results confirm that our framework provides superior modeling capacity for complex dynamics with varying temporal dependencies. The code is available at `https://github.com/cuiwjTech/NeurIPS2025_FADE`.

## 1 Introduction

Neural differential equations have emerged as powerful tools for modeling continuous-time dynamics in various machine learning tasks. By integrating neural networks with differential equations, these models can effectively learn the underlying dynamics of complex systems directly from data. The most representative model in this category is the integer-order neural Ordinary Differential Equation (ODE) [1–3], which formulates the evolution of hidden states as an integer-order ODE system. This approach has demonstrated success in various machine learning tasks, including time series forecasting [4–7], graph representation learning [8–10], physics modeling [11–13], adversarial robustness [14–16], and generative modeling [17, 18].

Despite their success, traditional neural ODEs based on integer-order differential equations exhibit inherent limitations in modeling long-range dependencies and memory effects. These stem from the Markovian nature of ODEs, where the future state depends solely on the current state rather than on

---

[*]First two authors contributed equally to this work.
[†]Correspondence to: Xuhao Li <lixh@ahu.edu.cn>.

39th Conference on Neural Information Processing Systems (NeurIPS 2025).

the trajectory's history [19]. Such locality constrains the model's ability to represent complex temporal patterns essential in many real-world tasks. To address these limitations, researchers have explored fractional-order differential equations (FDEs) as an alternative general mathematical framework. FDEs extend the classical derivative operator $\frac{d}{dt}$ to non-integer orders $\alpha$, denoted $\frac{d^\alpha}{dt^\alpha}$, thereby introducing a non-local memory term that naturally integrates past information. Thanks to this non-local property, FDE-based models can capture long-term dependencies, making them well-suited for systems with memory or long-range interactions. They have found applications across diverse fields, including anomalous diffusion processes [20], viscoelastic materials [21], electromagnetic wave propagation [22], and financial time series analysis [23]. In physics-informed machine learning, fractional physics-informed neural networks (fPINNs) [24] enforce underlying physical laws via FDEs, spawning further developments [25, 26]. Furthermore, recent works have incorporated FDEs into neural network architectures, giving rise to fractional neural ODE models [19, 27–30]. For instance, [19, 27] propose fractional graph neural ODE models with fractional diffusion and oscillator mechanisms to propagate information over graphs, demonstrating improved task performance and increased robustness compared to their integer-order counterparts [8, 9]. The work by [31] on generative fractional diffusion models has indicated that the fractional method can achieve greater pixel-level variation and overall image fidelity.

However, most existing fractional operators and derived neural models employ fixed fractional kernels that assign predefined weights to historical states, thereby limiting adaptability to diverse data patterns. For example, [19, 27, 29] use the Caputo fractional operator $D^\alpha$ with a fixed kernel $(t - \tau)^{-\alpha}$ (where $\alpha$ is the fractional derivative order), which applies a non-adaptive weighting to the trajectory. The main challenge in developing effective fractional neural models lies in designing appropriate kernel functions that can adapt to the complex dynamics of different systems. Fixed kernels with constant weights might not be optimal for capturing the varying importance of historical states across different features and time points. Several works have adopted more flexible frameworks. For example, the work [32] proposes the generalized $\psi$-Caputo fractional derivative with the kernel in the form of $(\psi(t) - \psi(\tau))^{-\alpha}$, where $\psi(\cdot)$ is an increasing continuously differentiable function. [33] proposes a neural variable-order FDE network using the variable-order Caputo fractional derivative $D_t^{\alpha(t, \mathbf{x}(\tau))}$. This approach assumes a kernel of the form $(t - \tau)^{-\alpha(t, \mathbf{x}(\tau))}$, where the derivative order $\alpha(t, \mathbf{x}(\tau))$ depends on hidden features $\mathbf{x}(\tau)$, capturing complex feature-updating dynamics with enhanced flexibility. However, those kernel functions still have limited capacity and does not account for correlations between historical states. The work [7] combines the modeling capabilities of integer-order neural ODEs with an attention mechanism applied to temporal partitions, capturing the dynamic changes of continuous-time systems. This is distinct from our research, wherein we focus on incorporating fractional operators for updating hidden features, modeled as a memory-inclusive dynamical process.

To overcome the limitations of fixed kernels, this paper introduces the generalized fractional attention differential equation (FADE). FADE generalizes fractional neural ODEs by incorporating learnable attention mechanisms [34, 35] directly into the fractional operator's kernel function. Instead of relying on predefined, static/uncorrelated weightings for historical states, FADE employs neural attention kernels that adaptively assign weights based on the contextual relevance of past information to the current system dynamics. This allows the model to learn feature-dependent representations of history, selectively emphasizing salient temporal dependencies while filtering less relevant information, leading to more expressive and flexible modeling.

**Main contributions.** This paper introduces a new continuous neural network framework based on generalized attention-based fractional operator kernels. Our key contributions include:

- We propose FADE, a novel fractional neural equation framework that integrates FDEs with learnable neural attention kernels. This enables adaptive, context-aware memory mechanisms that can selectively emphasize relevant historical dependencies while filtering less important temporal information, advancing beyond existing approaches with static/uncorrelated memory weightings.

- We provide rigorous theoretical foundations for FADE by analyzing essential kernel properties, including boundedness under both singular and nonsingular cases. We establish the well-posedness of the resulting neural integral equations using Banach fixed-point arguments, ensuring solution uniqueness under appropriate conditions. Additionally, we present a convergence analysis for the numerical discretization solver, ensuring the framework's practical implementability.

- We conduct extensive experiments on fluid flow, graph learning problems, spatio-temporal traffic forecasting, urban population mobility and biological neural spike trains tasks, demonstrating that FADE consistently outperforms integer-order neural ODE models and existing fractional approaches, confirming its superior capacity for modeling complex dynamics.

**Related Work.** Our work mainly focuses on Neural Differential Equations and neural network attention mechanisms. The related work is presented below. For a detailed discussion, please refer to Appendix A.

- **Neural Differential Equations** Neural Differential Equations (NDEs) unify neural networks and differential equations for modeling continuous dynamics. Neural ODEs (NODEs) [2] and their variants [36, 37] learn hidden state evolution by parameterizing ODEs. Yet, these use integer-order calculus, limiting their ability to capture memory effects and historical dependencies. To address this, FDE-based models [19, 33] have consequently been developed and widely applied. FROND [19] employs FDEs in graph learning, alleviating oversmoothing. Subsequently, NvoFDE [33] and DRAGON [38] were introduced to capture the distinct memory characteristics inherent in FDEs, but still rely on fixed kernels with static historical weights. Our work advances this by designing adaptive kernels within the FDE framework for richer spatio-temporal modeling.

- **Attention Mechanisms in Neural Networks** Attention mechanisms are central to modern deep learning, allowing models to focus on informative inputs. The Transformer [34] introduced self-attention, enabling efficient modeling of long-range dependencies across domains. In continuous-time models, attention mechanisms have been integrated with NDEs to enhance their expressiveness, allowing the model to selectively focus on relevant parts of the input sequence [39]. Attention mechanisms have also been applied to graph neural networks, resulting in Graph Attention Networks (GATs) [40]. However, the integration of attention with fractional-order models remains largely underexplored. Attention-based kernels in fractional NDEs offer a promising direction toward more flexible and adaptive modeling.

## 2 Preliminaries and Motivations

A distinctive strength of fractional calculus lies in its ability to model systems with non-local interactions, where the future state of a system depends significantly on its historical trajectory through specialized kernel functions. This section provides a concise overview of fractional calculus and illustrates its relationship to traditional calculus. Additionally, we present a brief introduction to both integer-order and fractional-order neural ODEs as foundational concepts for our work.

### 2.1 Fractional Calculus

We begin with a review of traditional calculus before introducing classical fractional-order integrals and derivatives. Additionally, we present the recently developed generalized fractional derivative formulation that incorporates a supplementary function $\psi(\cdot)$ within the kernel. For a more comprehensive exploration of fractional calculus theory, please see Appendix B.

#### 2.1.1 Traditional Calculus

Consider an $d$-dimensional function $\mathbf{x}(t) \in \mathbb{R}^d$ with respect to (w.r.t.) time $t$. The traditional first-order derivative, which quantifies the instantaneous rate of change, is defined as:

$$\frac{\mathrm{d}\mathbf{x}(t)}{\mathrm{d}t} := \lim_{\Delta t \to 0} \frac{\mathbf{x}(t + \Delta t) - \mathbf{x}(t)}{\Delta t}. \tag{1}$$

Let $J$ denote the classical integration operator over an interval $[a, b]$, defined as $J\mathbf{x}(t) := \int_a^t \mathbf{x}(\tau)\,\mathrm{d}\tau$. For any positive integer $m \in \mathbb{N}^+$, we define the iterated integral operator $J^m$ by $J^1 := J$ and $J^m := JJ^{m-1}$ for $m \geq 2$. Equivalently, applying integration by parts yields [41][Lemma 1.1.]:

$$J^m \mathbf{x}(t) = \frac{1}{(n-1)!} \int_a^t (t - \tau)^{m-1} \mathbf{x}(\tau)\,\mathrm{d}\tau \text{ with } m \in \mathbb{N}^+. \tag{2}$$

#### 2.1.2 Fractional Operators

The concepts of fractional-order integrals and derivatives generalize their integer-order counterparts. The commonly used Riemann-Liouville fractional integral [41], denoted by $^{\mathrm{RL}}J_{a+}^\alpha$ for a positive real

order $\alpha \in \mathbb{R}^+$, is defined as

$$^{\mathrm{RL}}J_{a+}^{\alpha}\mathbf{x}(t) := \frac{1}{\Gamma(\alpha)} \int_a^t (t-\tau)^{\alpha-1}\mathbf{x}(\tau)\,\mathrm{d}\tau, \tag{3}$$

where $\Gamma(\alpha)$ is the gamma function. Unlike the integer-order $m$ in traditional integrals, the order $\alpha$ can take any positive real value.

The literature offers various definitions for fractional integrals and derivatives, e.g., Riemann-Liouville, Hadamard, Erdélyi-Kober, and Caputo [42], which differ in how they weight historical trajectories. We mainly consider the Caputo fractional derivative because it preserves the same initial conditions as traditional integer-order differential equations. Throughout this work, we focus on the left-sided fractional derivative; the corresponding results for the right-sided derivative can be obtained analogously with appropriate modifications. We consider the case where the fractional order $\alpha \in (0,1]$, and the formulation for any $\alpha > 0$ is presented in Appendix B.

**Definition 1** (Classic Caputo Fractional Derivative). *The Caputo fractional derivative of order $\alpha \in (0,1]$ for a function $\mathbf{x}(t)$ over an interval $[a,b]$ is defined as follows [41]:*

$$^{\mathrm{C}}D_{a+}^{\alpha}\mathbf{x}(t) := \frac{1}{\Gamma(1-\alpha)} \int_a^t (t-\tau)^{-\alpha}\mathbf{x}'(\tau)\,\mathrm{d}\tau, \tag{4}$$

*where $\mathbf{x}'(\tau)$ is the first-order derivative of $\mathbf{x}(\tau)$.*

**Remark 1.** *When $\alpha = 1$, the Caputo fractional derivative $^{\mathrm{C}}D_{a+}^{\alpha}$ coincides with the standard first-order derivative $\frac{\mathrm{d}}{\mathrm{d}t}$ (see [41] (Theorem 7.1)). Therefore, (4) generalizes (1). It is evident from (4) that the fractional derivative incorporates the historical states of the function $\mathbf{x}(t)$ via the stationary power-law kernel $(t-\tau)^{-\alpha}$, highlighting its memory-dependent nature. In contrast, the integer-order derivative only represents the local rate of change of the function.*

**Definition 2** (Variable-Order Caputo Fractional Derivative). *When the fractional order $\alpha$ is allowed to vary with time $t$, the generalized variable-order Caputo fractional derivative of $\mathbf{x}(t)$ is defined as:*

$$^{\mathrm{C}}D_{a+}^{\alpha(t)}\mathbf{x}(t) := \frac{1}{\Gamma(1-\alpha(t))} \int_a^t (t-\tau)^{-\alpha(t)}\mathbf{x}'(\tau)\,\mathrm{d}\tau, \quad 0 < \alpha(t) \le 1. \tag{5}$$

**Remark 2.** *Compared to the classic Caputo fractional derivative (4), the variable-order derivative (5) employs a non-stationary power-law kernel. This allows it to dynamically adjust its memory structure, enabling the characterization of more complex processes than (4) [43, 44]. In more general settings, $\alpha(t)$ can be extended to depend on other parameters, e.g., $\alpha(t, \mathbf{x}(t))$.*

**Definition 3** ($\psi$-Caputo Fractional Derivative). *Let $\psi \in C^1([a,b])$ be a continuously differentiable scalar function such that $\psi(t)$ is increasing on $[a,b]$. The $\psi$-Caputo fractional derivative of $\mathbf{x}(t)$ of order $\alpha \in (0,1]$ is defined by [32]:*

$$^{\mathrm{C}}D_{a+}^{\alpha,\psi}\mathbf{x}(t) := \frac{1}{\Gamma(1-\alpha)} \int_a^t (\psi(t) - \psi(\tau))^{-\alpha}\mathbf{x}'(\tau)\,\mathrm{d}\tau. \tag{6}$$

**Remark 3.** *When $\psi(t) = t$, the $\psi$-Caputo fractional derivative (6) reduces to the classic Caputo fractional derivative (4). Therefore, the $\psi$-Caputo fractional derivative is also a natural generalization of (4). The main difference is that the $\psi$-Caputo definition utilizes a generalized kernel $(\psi(t) - \psi(\tau))^{-\alpha}$, where the function $\psi(\cdot)$ offers greater flexibility in weighting past values of the function.*

• **Observation and Motivation:** The preceding review of various Caputo fractional derivatives highlights their defining feature: the use of distinct weighting kernels, which can be static or designed to vary dynamically with time $t$. However, since these kernels depend only on $t$ and $\tau$ and not on past states $\mathbf{x}(\tau)$, they cannot adjust their weighting based on the states correlation in the trajectory. *In this paper, we propose to overcome this limitation by developing a more generalizable learnable attention kernel that extends beyond the capabilities of the above approaches, enabling memory weightings based on both temporal information and the contextual relationships between past and current states.*

## 2.2 Integer- and Fractional-Order Neural ODEs

In an integer-order neural ODE model, the process of transforming an initial feature vector $\mathbf{x}(0) = \mathbf{x}_0 \in \mathbb{R}^n$ into an output feature vector $\mathbf{x}(T) \in \mathbb{R}^n$ is dictated by the following first-order ODE:

$$\frac{\mathrm{d}\mathbf{x}(t)}{\mathrm{d}t} = f_{\boldsymbol{\theta}}(t, \mathbf{x}(t)) \tag{7}$$

In this equation, the neural network function $f_{\boldsymbol{\theta}}$, which is parameterized by $\boldsymbol{\theta}$ andmaps from $[0, \infty) \times \mathbb{R}^d$ to $\mathbb{R}^d$, defines the learnable dynamics responsible for updating the hidden features. The trajectory $\mathbf{x}(t)$ illustrates the continuous change of the system's hidden state over time.

When equation (7) is solved using the Euler discretization method, the update rule takes the form $\mathbf{x}(t + \Delta t) = \mathbf{x}(t) + \Delta t f_{\boldsymbol{\theta}}(t, \mathbf{x}(t))$. This structure shows a strong parallel to that of a ResNet with skip connections [45], where the variable $t$ can be seen as analogous to network depth or layer count. Furthermore, when addressing tasks with a temporal component, the dimension $t$ can directly correspond to the actual time dimension. In such cases, the neural ODE can be interpreted as a continuous-time recurrent neural network, as explored in studies such as [46–48].

Analogously, in a fractional-order neural ODE model [19, 29], feature dynamics are governed by:

$$^{\mathrm{C}}D_{a+}^{\alpha}\mathbf{x}(t) = f_{\boldsymbol{\theta}}(t, \mathbf{x}(t)), \quad 0 < \alpha \leq 1. \tag{8}$$

In this formulation, $f_{\boldsymbol{\theta}}$ characterizes the trainable fractional derivatives governing the hidden state dynamics. Starting from the initial condition $\mathbf{x}(0) = \mathbf{x}_0$, the system state $\mathbf{x}(t)$ evolves up to a predefined terminal time $T$. Computation of $\mathbf{x}(T)$ is accomplished using a forward fractional differential equation solver, such as the fractional explicit Adams–Bashforth–Moulton method [49]. Similar to integer-order models, the dimension $t$ can serve as a continuous analog to discrete layer indices or align directly with actual time dimensions in temporal tasks. However, rather than employing simple skip connections as in integer-order ODEs, fractional-order ODEs exhibit inherently dense connectivity patterns [50, 19].

**Remark 4.** *Both the integer- and fractional-order neural ODEs presented in* (7) *and* (8) *can be equivalently reformulated as integral equations, as we will demonstrate in Section 3.1. Motivated by this equivalence, we propose the generalized FADE framework, which not only encapsulates these formulations as special cases but also extends to scenarios where the fractional derivatives in* (8) *are replaced by variable-order or $\psi$-Caputo fractional derivatives shown in* (5) *and* (6). *This unified framework offers unprecedented flexibility in modeling complex continuous dynamics with adaptive memory mechanisms.*

## 3 Generalized Neural Fractional Attention Differential Equation

In this section, we introduce the FADE framework, a novel approach that generalizes fractional- and integer-order neural ODEs by incorporating learnable attention mechanisms directly into the fractional operator's kernel function. We first present the comprehensive framework formulation in Section 3.1, followed by a rigorous analysis of its theoretical properties in Section 3.2. Subsequently, we develop the numerical discretization solver necessary for implementing FADE and provide a detailed convergence analysis in Section 3.3. This systematic development establishes both the theoretical foundations and practical implementability of our proposed framework for modeling complex dynamics with adaptive memory mechanisms. All proofs are provided in Appendix C.

### 3.1 Framework

While it is possible to directly incorporate attention mechanisms that generalize the kernel functions presented in the Caputo fractional derivative definitions (4) to (6), we instead opt for a more elegant presentation by first transforming the integer-order and fractional-order neural ODEs in Section 2.2 into equivalent integration equations with different kernels. Specifically, the integer-order neural ODE model (7) can be equivalently expressed as:

$$\frac{\mathrm{d}\mathbf{x}(t)}{\mathrm{d}t} = f_{\boldsymbol{\theta}}(t, \mathbf{x}(t)) \iff \mathbf{x}(t) = \mathbf{x}(a) + \int_a^t f_{\boldsymbol{\theta}}(t, \mathbf{x}(\tau)) \, \mathrm{d}\tau, \tag{9}$$

with initial input $\mathbf{x}(a)$. Similarly, based on the relationship between fractional-order ODE and its integral form [41][Lemma 6.2.], we have:

$$^{\mathrm{C}}D_{a+}^{\alpha}\mathbf{x}(t) = f_{\boldsymbol{\theta}}(t, \mathbf{x}(t)) \iff \mathbf{x}(t) = \mathbf{x}(a) + \int_a^t \frac{(x-t)^{\alpha-1}}{\Gamma(\alpha)} f_{\boldsymbol{\theta}}(t, \mathbf{x}(\tau)) \, \mathrm{d}\tau. \tag{10}$$

The neural variable-order fractional differential equation networks presented in [33] adopt the following variable-order differential and integral equations:

$$^{\mathrm{C}}D_{a+}^{\alpha(t,\mathbf{x}(t))}\mathbf{x}(t) = f_{\boldsymbol{\theta}}(t, \mathbf{x}(t)) \text{ and } \mathbf{x}(t) = \mathbf{x}(a) + \int_a^t \frac{(t-\tau)^{\alpha(t,\mathbf{x}(t))-1}}{\Gamma(\alpha(t,\mathbf{x}(t)))} f_{\boldsymbol{\theta}}(t, \mathbf{x}(t)) \, \mathrm{d}\tau. \tag{11}$$

Furthermore, if we consider the $\psi$-Caputo fractional derivative (6) with $\psi(t)$ being a learnable function w.r.t. $t$, we can propose the following $\psi$-fractional-order differential and integral equations:

$$^{\mathrm{C}}D_{a+}^{\alpha,\psi}\mathbf{x}(t) = f_{\boldsymbol{\theta}}(t,\mathbf{x}(t)) \iff \mathbf{x}(t) = \mathbf{x}(a) + \int_a^t \frac{\psi'(t)(\psi(x)-\psi(t))^{\alpha-1}}{\Gamma(\alpha)} f_{\boldsymbol{\theta}}(t,\mathbf{x}(\tau))\,\mathrm{d}\tau, \quad (12)$$

where the equivalence can be established from [32].

To gain further insight, we rewrite the aforementioned integral equations in a unified formulation as:

$$\mathbf{x}(t) = \mathbf{x}(a) + \int_a^t K(t,\tau,\mathbf{x}(t)) f_{\boldsymbol{\theta}}(t,\mathbf{x}(\tau))\,\mathrm{d}\tau, \quad (13)$$

where $K(t,\tau,\mathbf{x})$ is a kernel function, with examples (9) to (12). This motivates us to study this integral equation and unveil its underlying relation to neural network architecture design. We find that the integral equation is essentially a continuous form of the following discrete iterative relation:

$$\mathbf{x}(t_n) = \mathbf{x}(t_0) + \sum_{j=0}^{n-1} \int_{t_j}^{t_{j+1}} K\left(t_n,t,\mathbf{x}(t_n)\right)\,\mathrm{d}\tau \times f_{\boldsymbol{\theta}}(t_j,\mathbf{x}(t_j)), \quad (14)$$

where $\{t_j\}_{j=0}^N$ is a grid of $[a,b]$ and $\mathbf{x}(t_j)$, $0 \le j \le n-1$, are sampled points from the trajectory $\mathbf{x}(t)$. It is observed that this discrete iterative relation shares a similar structure with the attention mechanism in Transformers [34] if we consider $\{\mathbf{x}(t_j)\}_{j=0}^{n-1}$ as input, $\{f_{\boldsymbol{\theta}}(t_j,\mathbf{x}(t_j))\}_{j=0}^{n-1}$ as values and $\int_{t_j}^{t_{j+1}} K(t_n,t,\mathbf{x}(t_n))\,\mathrm{d}\tau$ as weights derived from query and key.

From the discussions above, a natural relationship between the integral equations and the attention mechanism in Transformers is established. However, this direct analogy highlights several limitations. The kernels in (9) and (10) utilize fixed, non-learnable weights. While the kernel in (11) allows its fractional order $\alpha(t,\mathbf{x}(t))$ to respond to the current time t and state $\mathbf{x}(t)$, its structure does not inherently learn complex correlations between different states along the trajectory. Similarly, the kernel in (12), despite incorporating learnable temporal features through $\psi(t)$, primarily accounts for temporal differences and does not adapt its weighting based on the evolving state vectors (e.g., $\mathbf{x}(\tau)$ or $\mathbf{x}(t)$). These factors can potentially restrict their modeling capacity and performance in practical applications.

Inspired by this insight, we propose the FADE framework, which is based on the following neural integral equation featuring an attention kernel:

$$\mathbf{x}(t) = \mathbf{x}(a) + \int_a^t K(t,\tau,\mathbf{x}(t),\mathbf{x}(\tau)) f_{\boldsymbol{\theta}}(t,\mathbf{x}(\tau))\,\mathrm{d}\tau, \quad (15)$$

where $K(t,\tau,\mathbf{x}(t),\mathbf{x}(\tau)) \ge 0$. This formulation also corresponds to proper masking in self-attention layers in the Transformer decoder [34], ensuring that each step only attends to previous steps and thereby preserving the causal, auto-regressive nature of the integral equation.

**Remark 5.** *Instead of relying on static or uncorrelated weightings for historical states, FADE employs neural attention kernels that adaptively assign weights by assessing the contextual relevance of past information to the current system dynamics. Compared to the specific kernels presented in (9) to (12), FADE provides a unified formulation where the kernels are constructed based on both temporal information and the contextual relationships between past and current states. This approach allows the model to selectively emphasize salient temporal dependencies while filtering out less relevant information, leading to more expressive and flexible modeling capabilities than (9) to (12).*

### 3.1.1 Attention Kernel Examples

In contrast to the predefined kernels from (9) to (12), which possess limited adaptivity, we now introduce several flexible forms of the attention kernel $K(t,\tau,\mathbf{x}(t),\mathbf{x}(\tau))$ that explicitly leverage attention mechanisms.

**Scaled Dot-Product Attention:** Motivated by [34], the kernel $K$ can be set as

$$K(t,\tau,\mathbf{x}(t),\mathbf{x}(\tau)) = \sigma\left(\frac{\mathbf{x}(t)^\top \mathbf{x}(\tau)}{\sqrt{d}}\right) \quad (16)$$

where $d$ is the dimension of $\mathbf{x}(t)$. $\sigma$ is the softmax function. In the continuous setting, it implies a normalization of attention weights across the relevant range of $\tau$ for a given $t$. Specifically, if $s(t, \tau) = \frac{\mathbf{x}(t)^\top \mathbf{x}(\tau)}{\sqrt{d_x}}$ is the score, then $\sigma(s(t, \tau))$ could be interpreted as $\frac{\exp(s(t,\tau))}{\int_a^t \exp(s(t,s'))ds'}$ to ensure weights integrate to 1. In discretized numerical solvers, the softmax is applied over the set of sampled past states $\{\mathbf{x}(\tau_j)\}$.

**Fractional Attention Kernel:** Motivated by the structure of fractional integral operators seen in (10) to (12), the kernel $K$ can be set as

$$K(t, \tau, \mathbf{x}(t), \mathbf{x}(\tau)) = \frac{\psi'(\tau)(\psi(t) - \psi(\tau))^{\alpha-1}}{\Gamma(\alpha)} \tilde{K}(\mathbf{x}(t), \mathbf{x}(\tau)), \tag{17}$$

Here, the term $\frac{\psi'(\tau)(\psi(t)-\psi(\tau))^{\alpha-1}}{\Gamma(\alpha)}$ introduces a fractional memory component and can be regarded as positional encoding weighting. $\psi(t)$ is a learnable, monotonically increasing function of time, which can be interpreted as a reparameterization of the time axis, allowing the model to learn problem-specific memory decay rates. The parameter $\alpha$ represents the fractional order. $\tilde{K}(\mathbf{x}(t), \mathbf{x}(\tau))$ is itself an attention function that depends on the states $\mathbf{x}(t)$ and $\mathbf{x}(\tau)$. This $\tilde{K}$ can be, for instance, the scaled dot-product attention described above or an additive attention mechanism [51]. This hybrid kernel thus allows the model to leverage both the structured memory of fractional operators and the dynamic, input-dependent weighting of attention mechanisms.

**Position Augmented Attention Kernel:** A common method for incorporating positional information, introduced in the original Transformer [34], involves adding positional encodings directly to the input representations. For a given position $t$, the sinusoidal positional encoding vector $\mathrm{PE}(t)$ is defined in [34][Sec 3.5.]. In this approach, the state vectors $\mathbf{x}(t)$ and $\mathbf{x}(\tau)$ are first augmented as $\tilde{\mathbf{x}}(t) = \mathbf{x}(t) + \mathrm{PE}(t)$ and $\tilde{\mathbf{x}}(\tau) = \mathbf{x}(\tau) + \mathrm{PE}(\tau)$. They can then be projected using learnable linear transformation matrices, $\mathbf{W}_q$ for the query and $\mathbf{W}_k$ for the key. The attention kernel is computed as:

$$K(t, \tau, \mathbf{x}(t), \mathbf{x}(\tau)) = \sigma\left(\frac{(\mathbf{W}_q\tilde{\mathbf{x}}(t))^\top(\mathbf{W}_k\tilde{\mathbf{x}}(\tau))}{\sqrt{n}}\right). \tag{18}$$

The product of $\mathbf{W}_q$ and $\mathbf{W}_k$ can be initialized as a single learnable matrix in the implementation.

### 3.2  Well-posedness of FADE

We begin with properties of kernel integral. The attention kernel employed in this work depends explicitly on the system states $\mathbf{x}(t)$ and $\mathbf{x}(\tau)$. We analyze the properties of the integral operator:

$$I_K\mathbf{x}(t) := \int_a^t K(t, \tau, \mathbf{x}(t), \mathbf{x}(\tau))\mathbf{x}(\tau)\mathrm{d}\tau. \tag{19}$$

This integral encompasses several well-known aforementioned operators as special cases. These special cases result in linear, bounded operators that satisfy the semigroup property, crucial for deriving equivalent differential forms [41, 32]. Generally, the semigroup property does not hold for the attention kernels defined here, typically leading to nonlinear operators. Nonetheless, boundedness, which is essential for ensuring well-posedness, remains valid. Below, we discuss an important singular scenario and leave other interesting cases in Appendix D. That is, we foucs on the kernel that admits the decomposition

$$K(t, \tau, \mathbf{x}(t), \mathbf{x}(\tau)) = \frac{(\psi(t) - \psi(\tau))^{\alpha-1}\psi'(\tau)}{\Gamma(\alpha)} \tilde{K}(\mathbf{x}(t), \mathbf{x}(\tau)), \quad 0 < \alpha < 1.$$

**Lemma 1** (Boundedness of the Kernel Integral Operator (19)). *Suppose that $\psi(t)$ monotone increasing and $\tilde{K}$ continuous and bounded, then $I_K$ remains bounded and satisfies $\|I_K\mathbf{x}\| \leq C\|\mathbf{x}\|$.*

Now we are ready to consider well-posedness. Ensuring well-posedness (i.e., uniqueness and robustness) of neural integral equations is crucial for their practical reliability and theoretical robustness.

**Theorem 1** (Uniqueness and Stability). *Suppose that $\tilde{K}$ satisfying the Lipschitz condition, $f_\theta(\tau, \mathbf{x})$ is bounded and globally Lipschitz continuous in $\mathbf{x}$. The integral equation (15) admits a unique continuous solution in interval $[a, a + \epsilon]$ some $\epsilon > 0$. Moreover, we have*

$$\|\mathbf{x}(t) - \tilde{\mathbf{x}}(t)\|_2 \leq C\|\mathbf{x}(a) - \tilde{\mathbf{x}}(a)\|_2, \quad a < t \leq b,$$

*which implies that the solution is stable or robust with respect to the perturbation of initial data.*

### 3.3 Solving FADE

The integral equation (15) is a nonlinear equation which can be solved using linearized technique or iterative method. Here, we adopt the latter one. Taking $x = t_j$ and approximating the integral in (15) using the trapezoidal rule yields

$$\mathbf{x}\left(t_j\right) = \phi\left(\mathbf{x}\left(t_j\right)\right), \quad \phi\left(\mathbf{x}\left(t_j\right)\right) = \mathbf{x}\left(t_0\right) + \sum_{k=0}^{j-1} K\left(t_j, t_k, \mathbf{x}\left(t_j\right), \mathbf{x}\left(t_k\right)\right) f_{\boldsymbol{\theta}}\left(t_k, \mathbf{x}\left(t_k\right)\right) h. \quad (20)$$

For above nonlinear problems, we apply basic iteration method to solve it. The procedure of this method and related analysis are discussed below.

**Basic Iteration Method.** Given an initial guess $\mathbf{x}^{(0)}\left(t_j\right) = \mathbf{x}\left(t_{j-1}\right)$, we iteratively compute $\mathbf{x}^{(L)}\left(t_j\right) = \phi\left(\mathbf{x}^{(L-1)}\left(t_j\right)\right)$, where $L \geq 1$.

**Convergence Criterion.** The convergence of the above iterative method relies on the following condition: If $\phi : \mathbb{R}^d \to \mathbb{R}^d$ satisfies a Lipschitz condition with a Lipschitz constant $C < 1$, then the iterative methods converge for any initial guess $\mathbf{x}^{(0)} \in [a, b]^d$.

Given that the assumptions of Theorem 1 are satisfied with appropriate generic constant $C$, it can be directly verified that basic iteration method proposed here will converge.

**Convergence Rate.** Define the iteration error $e^{(L)} = \mathbf{x}^{(L)} - \mathbf{x}^{(L-1)}$. An iterative method has order of convergence $p$ if there exists a constant $C \neq 0$ such that $\lim_{L\to\infty} \left\| e^{(L+1)} \right\| / \left\| e^{(L)} \right\|^p = C$. For Basic Iteration method, applying the Lipschitz property of $\phi$, we obtain $\left\| e^{(L)} \right\| / \left\| e^{(L-1)} \right\| = \left\| \phi\left(\mathbf{x}^{(L-1)}\right) - \phi\left(\mathbf{x}^{(L-2)}\right) \right\| / \left\| \mathbf{x}^{(L-1)} - \mathbf{x}^{(L-2)} \right\| < 1$, indicating a linear convergence rate.

### 3.4 FADE Examples

FADE is a general framework that extends the applicability of both integer- and fractional-order models. This section mainly presents several FADE variants designed for graph learning tasks. The details of the new continuous models based on FADE are presented in Appendix E.

## 4 Experiments

We carry out extensive experiments to validate the performance of our proposed approach on fluid flow, node classification on graphs, traffic forecasting, urban population mobility and biological neural spike trains. Experiments for urban population mobility and biological neural spike train dynamics are shown in Appendix I. All implementations are developed using the PyTorch framework[52] on a single NVIDIA RTX4090 24GB GPU.

### 4.1 Fluid Flow Prediction

**Datasets and Methods**. We evaluated the proposed FADE on turbulent boundary-layer flow [53], with velocity fields measured using particle image velocimetry at five Reynolds numbers (Re = 600, 980, 1370, 1780, 2220), each containing approximately 6,000 snapshots. We trained on four Reynolds numbers and tested on the fifth. Models observed 6 consecutive snapshots to predict the next 6, using a CNN encoder–decoder for spatial feature extraction and LSTM, Transformer, Neural ODE, or our FADE for latent state temporal evolution. Turbulent flows exhibit strong memory effects due to the cascade of energy across different scales and the persistence of coherent structures. The non-local temporal dependencies in turbulence make it an ideal testbed for our attention-based fractional framework, which can adaptively weight historical flow states based on their relevance to current dynamics.

**Performance and Analysis**. The preliminary prediction results are summarized in Table 1, demonstrating that FADE's adaptive memory mechanism effectively captures the nonlinear multi-scale temporal dependencies inherent in turbulent flows, outperforming other approaches. The GPU memory usage of FADE is slightly higher than the baseline LSTM and ODE models, but comparable to the Transformer. This is expected since the extra computational complexity and memory usage from the attention mechanism are inevitable and standard. Importantly, with nearly the same GPU memory

Table 1: Average prediction error and memory consumption of different models on turbulent vector field prediction.

| Model | LSTM+CNN | Transformer+CNN | ODE+CNN | FADE+CNN (ours) |
|---|---|---|---|---|
| RMSE | 1.5035 | 0.4701 | 0.5962 | **0.3215** |
| MAE | 0.6027 | 0.3532 | 0.2799 | **0.2041** |
| Training Memory | 11372MiB | 14142MiB | 11250MiB | 14722MiB |

usage, FADE performs much better than the Transformer. In fact, due to the flexibility of FADE, one can incorporate efficient attention mechanisms such as [54] to alleviate GPU memory usage, which can reduce the computational cost from $O\left(L^2\right)$ to $O(L \log L)$ in practice.

## 4.2 Node Classification on Graph

**Datasets and Methods**. We evaluate our approach on a variety of datasets with different graph structures. For the Airport dataset, we adopt preprocessing protocols from [55]. For the others, we follow the protocols from GRAND [56], specifically using random splits applied to the largest connected component. We compare our method against several established GNN models, including GCN [57], HGCN [55], GIL [58], GRAND [56], FROND [19], DRAGON [38] and NvoFDE [33], which are detailed in Appendix F. GRAND, FROND, DRAGON, and NvoFDE adopt specific kernel functions as listed in (9) to (12). We primarily utilize the Fractional Attention Kernel presented in Section 3.1.1. We examine two variants of FADE: FADE-l and FADE-nl provided in Appendix E. Taking the Cora dataset as an example to illustrate the experimental parameter settings, we set time = 25, step size = 1, learning rate = 0.01, weight decay = 0.05, epoch = 800 and dim = 256. For more details regarding the dataset and extended experimental results, please refer to Appendix G.

Table 2: Node classification results (%) for random train-val-test splits. The best result is highlighted in red.

| Model | Cora | Citeseer | Pubmed | CoauthorCS | Computer | Photo | CoauthorPhy | ogbn-arxiv | Airport |
|---|---|---|---|---|---|---|---|---|---|
| GCN[57] | 81.5±1.3 | 71.9±1.9 | 77.8±2.9 | 91.1±0.5 | 82.6±2.4 | 91.2±1.2 | 92.8±1.0 | 72.2±0.3 | 81.6±0.6 |
| HGCN[55] | 78.7±1.0 | 65.8±2.0 | 76.4±0.8 | 90.6±0.3 | 80.6±1.8 | 88.2±1.4 | 90.8±1.5 | 59.6±0.4 | 85.4±0.7 |
| GIL[58] | 82.1±1.1 | 71.1±1.2 | 77.8±0.6 | 89.4±1.5 | – | 89.6±1.3 | – | – | 91.5±1.7 |
| GRAND-l[56] | 83.6±1.0 | 73.4±0.5 | 78.8±1.7 | 92.9±0.4 | 83.7±1.2 | 92.3±0.9 | 93.5±0.9 | 71.9±0.2 | 80.5±9.6 |
| F-GRAND-l[19] | 84.8±1.1 | 74.0±1.5 | 79.4±1.5 | 93.0±0.3 | 84.4±1.5 | 92.8±0.6 | 94.5±0.4 | **72.6±0.1** | 98.1±0.2 |
| D-GRAND-l[38] | 85.1±1.3 | 74.5±1.1 | 79.6±2.6 | 93.2±0.3 | 87.3±1.3 | 93.1±0.8 | 94.6±0.2 | – | 98.5±0.1 |
| Nvo-GRAND-l[33] | 86.0±0.5 | 75.6±0.8 | **80.8±1.2** | 93.4±0.2 | 87.9±0.8 | 94.1±0.2 | 94.7±0.2 | 71.8±0.1 | 98.7±0.2 |
| FADE-l (ours) | **86.4±0.5** | 76.1±0.6 | 80.7±0.7 | **93.5±0.1** | **88.3±0.9** | **94.4±0.2** | 94.7±0.2 | 72.0±0.2 | **98.8±0.1** |
| GRAND-nl[56] | 82.3±1.6 | 70.9±1.0 | 77.5±1.8 | 92.4±0.3 | 82.4±2.1 | 92.4±0.8 | 91.4±1.3 | 71.2±0.2 | 90.9±1.6 |
| F-GRAND-nl[19] | 83.2±1.1 | 74.7±1.9 | 79.2±0.7 | 92.9±0.4 | 84.1±0.9 | 93.1±0.9 | 93.9±0.5 | 71.4±0.3 | 96.1±0.7 |
| D-GRAND-nl[38] | 83.9±1.3 | 74.8±1.6 | 79.5±2.6 | 93.1±0.3 | 87.1±1.0 | 93.4±0.5 | 94.3±0.6 | – | 97.7±0.4 |
| Nvo-GRAND-nl[33] | 85.4±1.0 | 75.9±0.6 | 80.6±0.7 | 93.4±0.2 | 87.2±1.4 | 94.0±0.3 | 94.6±0.2 | 72.0±0.2 | 98.4±0.2 |
| FADE-nl (ours) | 86.0±0.4 | **76.2±0.8** | 80.6±0.7 | 93.4±0.2 | 87.7±0.9 | 94.1±0.3 | **94.8±0.2** | 72.0±0.1 | 98.6±0.1 |

**Performance and Analysis**. The results presented in Table 2 show that our FADE model achieves the best results on most datasets. Compared to GRAND, F-GRAND and D-GRAND models, FADE demonstrates superior performance. For example, on Cora, Pubmed and Photo datasets, FADE-l shows performance gains of 1.3%–2.8%, 1.1%–2.0%, and 1.1%–2.1%, respectively, while FADE-nl achieves improvements of 2.1%–3.7%, 1.1%–3.1%, and 0.7%–1.7%, respectively. Against Nvo-GRAND, both FADE variants consistently deliver competitive or better results across all datasets. These results highlight the strong performance advantage of FADE, demonstrating its ability to effectively capture complex graph dynamics.

## 4.3 Traffic Forecasting

**Datasets and Methods**. To further validate the effectiveness of the FADE model, we conduct experiments on time-series forecasting tasks using four real-world traffic datasets: PeMSD7(M), PeMSD7(L), PeMS04, and PeMS08. These datasets are collected in real time every 30 seconds by the Caltrans Performance Measurement System (PeMS) [59]. A more detailed description can be found in Appendix H. Maintaining consistency with [4], we train all datasets for 200 epochs

using the Adam optimizer and a batch size of 64. An early stopping strategy is applied, with a patience of 15 iterations on the validation dataset. For performance evaluation, we employ three widely-used metrics: Mean Absolute Error (MAE), Root Mean Square Error (RMSE), and Mean Absolute Percentage Error (MAPE), which collectively provide a comprehensive assessment of the prediction accuracy. For comparative analysis, we select several baseline models, including HA[60], VAR[60], TCN[61], DSANet[62], AGCRN[63], STFGNN[64], Z-GCNETs[65], STGODE[6] and STG-NCDE[4]. Following [4], we refer to our model as spatio-temporal FADE (STG-FADE) in this part.

**Performance and Analysis**. The results presented in Table 3 reveal that the proposed STG-FADE generally outperforms other models in forecasting accuracy across the PeMSD4, PeMSD8, PeMSD7(M), and PeMSD7(L) datasets. STG-FADE achieves the lowest MAE and RMSE values on PeMSD4, PeMSD8, and PeMSD7(M) alongside competitive MAPE performance. On PeMSD7(L), STG-FADE shows excellent results in all three metrics with the best MAPE value, highlighting its generalization on large-scale traffic datasets. More experimental results are presented in Appendix H. Overall, STG-FADE exhibits superior and stable performance, effectively capturing the spatio-temporal information inherent in traffic forecasting.

Table 3: Forecasting error on PeMSD4, PeMSD8, PeMSD7(M), and PeMSD7(L). The best and the second-best results are highlighted in red and blue, respectively.

| Model | PeMSD4 | | | PeMSD8 | | | PeMSD7(M) | | | PeMSD7(L) | | |
|---|---|---|---|---|---|---|---|---|---|---|---|---|
| | MAE | RMSE | MAPE | MAE | RMSE | MAPE | MAE | RMSE | MAPE | MAE | RMSE | MAPE |
| HA[60] | 38.03 | 59.24 | 27.88% | 34.86 | 59.24 | 27.88% | 4.59 | 8.63 | 14.35% | 4.84 | 9.03 | 14.90% |
| VAR[60] | 24.54 | 38.61 | 17.24 % | 19.19 | 29.81 | 13.10% | 4.25 | 7.61 | 10.28% | 4.45 | 8.09 | 11.62% |
| TCN[61] | 23.22 | 37.26 | 15.59% | 22.72 | 35.79 | 14.03% | 4.36 | 7.20 | 9.71% | 4.05 | 7.29 | 10.43% |
| DSANet[62] | 22.79 | 35.77 | 16.03% | 17.14 | 26.96 | 11.32% | 3.52 | 6.98 | 8.78% | 3.66 | 7.20 | 9.02% |
| AGCRN[63] | 19.83 | 32.26 | 12.97% | 15.95 | 25.22 | 10.09% | 2.79 | 5.54 | 7.02% | 2.99 | 5.92 | 7.59% |
| STFGNN[64] | 20.48 | 32.51 | 16.77% | 16.94 | 26.25 | 10.60% | 2.90 | 5.79 | 7.23% | 2.99 | 5.91 | 7.69% |
| Z-GCNETs[65] | 19.50 | 31.61 | 12.78% | 15.76 | 25.11 | 10.01% | 2.75 | 5.62 | 6.89% | 2.91 | 5.83 | 7.33% |
| STGODE[6] | 20.84 | 32.82 | 13.77% | 16.81 | 25.97 | 10.62% | 2.97 | 5.66 | 7.36% | 3.22 | 5.98 | 7.94% |
| STG-NCDE[4] | 19.21 | 31.09 | 12.76% | 15.45 | 24.81 | 9.92% | 2.68 | 5.39 | 6.76% | 2.87 | 5.76 | 7.31% |
| STG-FADE (ours) | 19.17 | 31.01 | 12.77% | 15.29 | 24.67 | 10.12% | 2.69 | 5.39 | 6.71% | 2.91 | 5.81 | 7.20% |

## 5    Conclusion

In this paper, we propose the generalized neural fractional attention differential equation (FADE), a novel continuous neural network framework that integrates fractional calculus with learnable neural attention kernels. By replacing the fixed kernel with adaptive attention, FADE effectively emphasizes relevant historical dependencies while filtering out less important information. Our analysis establishes boundedness, well-posedness, and convergence, ensuring the soundness of the proposed framework. Extensive experiments confirm FADE superior performance and stability. Overall, FADE advances continuous-depth learning by combining fractional calculus long-memory modeling with adaptive attention, enabling effective learning from complex temporal data.

## Acknowledgments and Disclosure of Funding

This work is supported by the National Natural Science Foundation of China under Grants 62225207, 62436008, 62576326, 12301491, U2268203, 62433005, 62272036, 62173167. To improve the readability, parts of this paper have been grammatically revised using ChatGPT [66].

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

## Introduction

This supplementary document provides extended explanations and additional results that support the claims presented in the main paper. The content is organized as follows.

1. A comprehensive related work is detailed in Appendix A.
2. Preliminaries on fractional calculus are presented in Appendix B.
3. All theoretical proofs in this paper are presented in Appendix C.
4. Nonsingular scenario for the kernel is shown in Appendix D.
5. FADE examples are presented in Appendix E.
6. Graph differential equation models are given in Appendix F.
7. Datasets and more experiments on the graph are available in Appendix G.
8. Datasets and more experiments for traffic forecasting are discussed in Appendix H.
9. Experiments for urban population mobility and biological neural spike train dynamics are provided in Appendix I.
10. Limitations and broader impact are discussed in Appendix J.

## A   Related Work

Our work builds on research in fractional differential equations, neural integer-order and fractional-order ODE models, and neural network attention mechanisms. We will present the related work from the following aspects.

### A.1   Fractional Differential Equations

Fractional Differential Equations (FDEs) generalize classical differential equations by allowing the order of differentiation to be a non-integer, thereby providing a powerful framework for modeling systems with memory and hereditary properties. The mathematical foundations of FDEs have been rigorously studied, with seminal contributions from [67–69]. Among the most prominent formulations are the Riemann–Liouville and Caputo derivatives [70], both of which use power-law kernels to encode memory effects. However, such power-law kernels often impose restrictions when modeling systems with heterogeneous or scale-dependent dynamics. To address these limitations, alternative definitions have been proposed, such as the Caputo–Fabrizio derivative with an exponential decay kernel [68], and the Atangana–Baleanu derivative with a Mittag-Leffler-based kernel [71]. These generalizations retain the core non-local structure of fractional calculus while enhancing modeling flexibility. More recently, variable-order fractional derivatives have attracted considerable attention due to their ability to reflect more flexible and complex dynamic memory mechanism in real-world phenomena [72, 43]. In these systems, the fractional order $\alpha$ is time-dependent, denoted as $\alpha(t)$, enabling a more precise representation of evolving dynamic behaviors.

In addition to the rich theoretical results, FDEs have found broad applicability across a wide range of fields. For instance, the authors offered a foundational overview of its practical uses in areas such as signal processing, system modeling and automatic control [73]. The authors showed that fractional calculus has emerged as a powerful mathematical framework for modeling complex systems in traffic forecasting [74]. The authors highlighted its role in viscoelasticity, illustrating how fractional-order models effectively capture the memory and hereditary characteristics inherent in such materials [75]. Meanwhile, FDEs have also been widely utilized to improve the performance of graph neural networks [76, 19, 33]. Despite considerable progress in both theory and applications, the design of kernel functions for fractional calculus remains largely underexplored. Particularly in neural differential equations, adaptive kernel functions can learn to assign appropriate weights to historical states based on relevance, which is still in infancy.

### A.2   Neural Differential Equations

Neural Differential Equations (NDEs) offer a unified framework that integrates neural networks with differential equations to model continuous-time dynamic systems, bridging the gap between

deep learning and classical dynamical systems. Among the various types of NDEs, Neural Ordinary Differential Equations (NODEs) and Neural Fractional Differential Equations (NFDEs) are more closely related to our work. Introduced by [2], NODEs model the evolution of hidden states by parameterizing the right-hand side of an ODE using neural networks. This model allows for adaptive computation in continuous time, offering memory efficiency and interpretability. To enhance the expressiveness of NODEs, several variants have been proposed. Augmented NODEs [36] expand the hidden state space with additional dimensions to improve the representational power and alleviate topological constraints that limit standard NODEs. Neural Controlled Differential Equations (NCDEs) [48] further generalize the NODEs framework to handle controlled systems in modeling irregular time series data, such as in finance or healthcare. Graph NODEs [37] have demonstrated strong performance on graph-structured data by integrating graph convolutional operations with continuous dynamics. Spatio-temporal graph NCDE [4] achieves significant performance improvements in traffic forecasting by integrating two NCDEs for temporal and spatial processing.

Despite their flexibility, NODEs-based models are limited by integer-order calculus, restricting their ability to capture memory and long-range dependencies, thus motivating interest in fractional-order extensions. Inspired by Graph NODEs, the FROND framework introduces a generalized fractional-order continuous GNN model using Caputo derivatives to capture non-local, memory-dependent dynamics, offering improved performance and mitigating oversmoothing in graph learning tasks [19]. Then, the DRAGON framework is proposed, which shows a distributed-order fractional continuous GNN that learns a superposition of derivative orders, enabling flexible and non-Markovian feature updating dynamics [27]. Recently, the NvoFDE framework introduces variable-order fractional differential operators into neural networks, enabling learnable and adaptive derivative orders based on time and hidden features [33]. However, most existing fractional neural models use fixed kernels with predefined weights assigned to historical states, limiting their flexibility and adaptability. Our work addresses this gap by introducing adaptive kernel functions into the fractional differential equation framework for improved temporal representation.

### A.3   Attention Mechanisms in Neural Networks

Attention mechanisms have emerged as a fundamental component of modern deep learning architectures, enabling models to dynamically prioritize informative parts of the input. The transformer architecture [34] introduces self-attention, which computes pairwise interactions between elements in a sequence, allowing for efficient modeling of long-range dependencies. This innovation has had transformative impacts across a variety of domains, including natural language processing, computer vision and time-series forecasting [77, 78]. In continuous-time systems, attention mechanisms have been integrated into neural differential equations to enhance the representational power. For example, Continuous Self-Attention Neural ODEs [39] extend Neural ODEs framework by integrating a lightweight self-attention mechanism, resulting in more flexible and interpretable dynamics. Similarly, attention-based Neural ODEs have been employed in spatio-temporal prediction tasks [79].

In Graph Neural Networks (GNNs), attention mechanisms have unlocked unprecedented flexibility in neighbor weighting and hierarchical feature propagation. Graph Attention Networks (GATs) [40] use self-attention to assign adaptive weights to neighboring nodes during aggregation, improving performance in scenarios with heterogeneous node importance. Extensions of GATs, such as multi-head and hierarchical attention models [80, 81], further enhance the model's ability to capture structural nuances. Despite these advances, the integration of attention with fractional-order models remains largely underexplored. Fractional calculus, known for its inherent memory and non-local properties, offers a natural framework to capture long-range dependencies. Combining it with attention mechanisms could lead to a new class of flexible and adaptive neural operators.

## B   Preliminaries on Fractional Calculus

This section offers additional material on fractional calculus theory, with key details presented in the main text of Section 2. Different from [38], we present results in terms of $\psi$-fractional derivative that is quite general than previous work ($\psi(t) = t$). For more detailed information, please refer to [32, 41]. We begin with the basic definitions.

**Definition 4** ($\psi$-Caputo Fractional Derivative). *Let $\alpha > 0$, $n \in \mathbb{N}$, and $I$ be an interval such that $-\infty \leq a < b \leq \infty$. Let $f, \psi \in C^n(I)$ be two functions such that $\psi$ is increasing and $\psi'(x) \neq 0$ for*

*all $x \in I$. The left $\psi$-Caputo fractional derivative of $f$ of order $\alpha$ is defined by*

$$^{C}D_{a+}^{\alpha,\psi}f(x) := \frac{1}{\Gamma(n-\alpha)} \int_a^x \psi'(t)(\psi(x) - \psi(t))^{n-\alpha-1} f_\psi^{[n]}(t)\, dt,$$

*and the right $\psi$-Caputo fractional derivative is given by*

$$^{C}D_{b-}^{\alpha,\psi}f(x) := \frac{(-1)^n}{\Gamma(n-\alpha)} \int_x^b \psi'(t)(\psi(t) - \psi(x))^{n-\alpha-1} f_\psi^{[n]}(t)\, dt,$$

*where $f_\psi^{[n]}(t) := \left(\frac{1}{\psi'(t)} \frac{d}{dt}\right)^n f(t)$, $n = [\alpha] + 1$ if $\alpha \notin \mathbb{N}$, and $n = \alpha$ if $\alpha \in \mathbb{N}$.*

*For $\alpha \in (0, 1)$, the left and right $\psi$-Caputo fractional derivatives reduce to*

$$^{C}D_{a+}^{\alpha,\psi}f(x) = \frac{1}{\Gamma(1-\alpha)} \int_a^x (\psi(x) - \psi(t))^{-\alpha} f'(t)\, dt,$$

*and*

$$^{C}D_{b-}^{\alpha,\psi}f(x) = \frac{-1}{\Gamma(1-\alpha)} \int_a^x (\psi(x) - \psi(t))^{-\alpha} f'(t)\, dt,$$

*respectively. For specific choices of the function $\psi$, the $\psi$-Caputo fractional derivative reduces to several well-known operators [82]. Throughout this work, we focus on the left-sided fractional derivative. The corresponding results for the right-sided derivative can be obtained analogously with appropriate modifications.*

To get some intuition, we provide a specific example below.

**Lemma 2.** *Given $\beta \in \mathbb{R}$ with $\beta > n$, consider the following function:*

$$f(x) = (\psi(x) - \psi(a))^{\beta-1}, \quad g(x) = (\psi(b) - \psi(x))^{\beta-1}.$$

*For $\alpha > 0$, we have:*

$$^{C}D_{a+}^{\alpha,\psi}f(x) = \frac{\Gamma(\beta)}{\Gamma(\beta-\alpha)}(\psi(x) - \psi(a))^{\beta-\alpha-1},$$

$$^{C}D_{b-}^{\alpha,\psi}g(x) = \frac{\Gamma(\beta)}{\Gamma(\beta-\alpha)}(\psi(b) - \psi(x))^{\beta-\alpha-1}.$$

Now we present the relation between fractional order derivative and integer order counterparts. This can be readily seen from the following theorem that is derived mainly using integration by parts.

**Theorem 2.** *Suppose that $f, \psi \in C^{n+1}[a, b]$. Then, for all $\alpha > 0$,*

$$^{C}D_{a+}^{\alpha,\psi}f(x) = \frac{(\psi(x) - \psi(a))^{n-\alpha}}{\Gamma(n+1-\alpha)} f_\psi^{[n]}(a) + \frac{1}{\Gamma(n+1-\alpha)} \int_a^x (\psi(x) - \psi(t))^{n-\alpha} \frac{d}{dt} f_\psi^{[n]}(t)\, dt,$$

*and*

$$^{C}D_{b-}^{\alpha,\psi}f(x) = (-1)^n \frac{(\psi(b) - \psi(x))^{n-\alpha}}{\Gamma(n+1-\alpha)} f_\psi^{[n]}(b)$$

$$- \frac{1}{\Gamma(n+1-\alpha)} \int_x^b (\psi(t) - \psi(x))^{n-\alpha} (-1)^n \frac{d}{dt} f_\psi^{[n]}(t)\, dt.$$

From this theorem, it is found that

$$\lim_{\alpha \to n^-} {}^{C}D_{a+}^{\alpha,\psi}f(x) = f_\psi^{[n]}(t).$$

We next present the relation between integration and differentiation of $\psi$-Caputo fractional function that is vital for the equivalent transformation between differential form and its integral form.

**Theorem 3.** *Given a function $f \in C^n[a, b]$ and $\alpha > 0$, we have:*

$$I_{a+}^{\alpha,\psi} \left( {}^C D_{a+}^{\alpha,\psi} f(x) \right) = f(x) - \sum_{k=0}^{n-1} \frac{f_\psi^{[k]}(a)}{k!} (\psi(x) - \psi(a))^k,$$

$$I_{b-}^{\alpha,\psi} \left( {}^C D_{b-}^{\alpha,\psi} f(x) \right) = f(x) - \sum_{k=0}^{n-1} \frac{(-1)^k f_\psi^{[k]}(b)}{k!} (\psi(b) - \psi(x))^k.$$

**Theorem 4.** *Given a function $f \in C^1[a, b]$ and $\alpha > 0$, we have*

$${}^C D_{a+}^{\alpha,\psi} I_{a+}^{\alpha,\psi} f(x) = f(x) \quad and \quad {}^C D_{b-}^{\alpha,\psi} I_{b-}^{\alpha,\psi} f(x) = f(x).$$

Obviously, one can simply apply $I_{a+}^{\alpha,\psi}$ on both sides of the differential equations to get its integral form. On the other hand, we can apply ${}^C D_{a+}^{\alpha,\psi}$ directly to integral equations in order to recover differential equations.

Lastly, we will show the semigroup law for the $\psi$-Caputo fractional derivative. Similar to classical fractional derivative [41], semigroup law does not hold in general for fractional derivative but it is indeed true for integrals. In what follows, we present a case that allows semigroup law.

**Theorem 5.** *If $f \in C^{m+n}[a, b]$ for some $m \in \mathbb{N}$ and $\alpha > 0$, then for all $k \in \mathbb{N}$ we have*

$$\left( I_{a+}^{\alpha,\psi} \right)^k \left( {}^C D_{a+}^{\alpha,\psi} \right)^m f(x) = \left( {}^C D_{a+}^{\alpha,\psi} \right)^m f(c) \cdot \frac{(\psi(x) - \psi(a))^{k\alpha}}{\Gamma(k\alpha + 1)},$$

$$\left( I_{b-}^{\alpha,\psi} \right)^k \left( {}^C D_{b-}^{\alpha,\psi} \right)^m f(x) = \left( {}^C D_{b-}^{\alpha,\psi} \right)^m f(d) \cdot \frac{(\psi(b) - \psi(x))^{k\alpha}}{\Gamma(k\alpha + 1)},$$

*for some $c \in (a, x)$ and $d \in (x, b)$.*

## C   All theoretical proofs

**Proof of Lemma 1.** Since

$$\|I_K \mathbf{x}\| \le C \int_a^t \frac{(\psi(t) - \psi(\tau))^{\alpha-1} \psi'(\tau)}{\Gamma(\alpha)} \|\mathbf{x}\| \mathrm{d}\tau \le C \frac{(\psi(t) - \psi(a))^\alpha}{\Gamma(1 + \alpha)} \|\mathbf{x}\|,$$

it is immediately seen that this operator is bounded.

**Proof of Theorem 1.** We shall prove the uniqueness as well as stability. To show the uniqueness, we define the operator as

$$T[\mathbf{x}](t) = \mathbf{x}(a) + \int_a^t \frac{(\psi(t) - \psi(\tau))^{\alpha-1} \psi'(\tau)}{\Gamma(\alpha)} \tilde{K}(\mathbf{x}(t), \mathbf{x}(\tau)) f_\theta(\tau, \mathbf{x}(\tau)) \mathrm{d}\tau,$$

The Lipschitz properties of $\tilde{K}$ and $f$ lead to:

$$\|T[\mathbf{x}_1] - T[\mathbf{x}_2]\| \le C \frac{(\psi(t) - \psi(a))^\alpha}{\Gamma(1 + \alpha)} \|\mathbf{x}_1 - \mathbf{x}_2\|,$$

Again, selecting a suitable $\epsilon > 0$ and invoking the Banach fixed-point theorem ensures a unique solution. To prove the stability, from (15), we shall get

$$\|\mathbf{x}(t) - \tilde{\mathbf{x}}(t)\|_2 \le \|\mathbf{x}(a) - \tilde{\mathbf{x}}(a)\|_2 + C \int_a^t \frac{(\psi(t) - \psi(\tau))^{\alpha-1} \psi'(\tau)}{\Gamma(\alpha)} \|\mathbf{x}(\tau) - \tilde{\mathbf{x}}(\tau)\|_2 d\tau.$$

Applying fractional Grönwall inequality [41, Lemma 6.19], we derive the stability result.

• **Observation and Motivation:** The preceding review of various Caputo fractional derivatives highlights their defining feature: the use of distinct weighting kernels, which can be static or designed to vary dynamically with time $t$. For high-dimensional states $\mathbf{x}(t)$, we can extend this idea by introducing a learnable vector $\boldsymbol{\psi}(t) = (\psi_1(t), \dots, \psi_n(t))$, where each $\psi_i(t)$ defines a component-wise kernel in a $\psi$-Caputo framework, enabling adaptive, dimension-specific memory modeling. However, since these kernels depend only on $t$ and $\tau$ and not on past states $\mathbf{x}(\tau)$ or the current state $\mathbf{x}(t)$, they cannot adjust their weighting based on the states correlation in the trajectory. *In this paper, we propose to overcome this limitation by developing a more generalizable learnable attention kernel that extends beyond the capabilities of the $\psi$-based approach. Our framework will incorporate mechanisms that can adapt memory weightings based on both temporal information and the contextual relationships between past and current states.*

## C.1 Solving FADE

The integral equation (15) is a nonlinear equation which can be solved using linearized technique or iterative method. Here, we adopt the latter one. Taking $x = t_j$ and approximating the integral in (15) using the trapezoidal rule yields

$$\mathbf{x}(t_j) = \mathbf{x}(t_0) + \sum_{k=0}^{j-1} K(t_j, t_k, \mathbf{x}(t_j), \mathbf{x}(t_k)) f_{\boldsymbol{\theta}}(t_k, \mathbf{x}(t_k)) h. \tag{21}$$

The iterative method for above nonlinear problems works as follows: taking the initial guess $\mathbf{x}(t_j^{(0)}) = \mathbf{x}(t_{j-1})$, then we conduct iterations based on

$$\mathbf{x}(t_j^{(L)}) = \mathbf{x}(t_0) + \sum_{k=0}^{j-1} K(t_j, t_k, \mathbf{x}(t_j^{(L-1)}), \mathbf{x}(t_k)) f_{\boldsymbol{\theta}}(t_k, \mathbf{x}(t_k)) h, \qquad L \geq 1.$$

This procedure will lead to a good approximation of $\mathbf{x}(t_j)$ after a few iterations. To address it, define

$$\mathbf{x}(t_j) = \phi(\mathbf{x}(t_j)), \quad \text{where} \quad \phi(\mathbf{x}(t_j)) = \mathbf{x}(t_0) + \sum_{k=0}^{j-1} K(t_j, t_k, \mathbf{x}(t_j), \mathbf{x}(t_k)) f_{\boldsymbol{\theta}}(t_k, \mathbf{x}(t_k)) h,$$

for the sake of simplicity. The proposed iterative solution to the discretized equation can formulated as the Basic Iteration method.

**Basic Iteration Method.** Given an initial guess $\mathbf{x}^{(0)}(t_j) = \mathbf{x}(t_{j-1})$, we iteratively compute

$$\mathbf{x}^{(L)}(t_j) = \phi\left(\mathbf{x}^{(L-1)}(t_j)\right), \quad L \geq 1.$$

### C.1.1 Convergence Criterion

The convergence of the above iterative method relies on the following condition: If $\phi : \mathbb{R}^d \to \mathbb{R}^d$ satisfies a Lipschitz condition with a Lipschitz constant $C < 1$, namely,

$$\|\phi(\mathbf{x}) - \phi(\mathbf{y})\| \leq C\|\mathbf{x} - \mathbf{y}\|, \quad \forall \mathbf{x}, \mathbf{y} \in [a, b]^d,$$

then the iterative methods converge for any initial guess $\mathbf{x}^{(0)} \in [a, b]^d$.

Given that the assumptions of Theorem 1 are satisfied with appropriate generic constant $C$, it can be directly verified that basic iteration method proposed here will converge. The essence is to make sure that the constant is less than 1.

### C.1.2 Convergence Rate

Define the iteration error $e^{(L)} = \mathbf{x}^{(L)} - \mathbf{x}^{(L-1)}$. An iterative method has order of convergence $p$ if there exists a nonzero constant $C$ such that

$$\lim_{L \to \infty} \frac{\left\|e^{(L+1)}\right\|}{\left\|e^{(L)}\right\|^p} = C.$$

For the Basic Iteration method, applying the Lipschitz property of $\phi$, we obtain

$$\frac{\left\|e^{(L)}\right\|}{\left\|e^{(L-1)}\right\|} = \frac{\left\|\phi\left(\mathbf{x}^{(L-1)}\right) - \phi\left(\mathbf{x}^{(L-2)}\right)\right\|}{\left\|\mathbf{x}^{(L-1)} - \mathbf{x}^{(L-2)}\right\|} \leq C < 1,$$

indicating a linear (first-order) convergence rate.

## D  Nonsingular scenario for the Kernel

Suppose that the kernal $K$ is bounded, Lipschitz continuous with respect to the last two variables. We find that the operator is bounded that is immediately seen from the boundedness of $K$. Besides, the integral equation is well-posed. The proof is as follows:

**Uniqueness.** Define operator:

$$T[\mathbf{x}](t) = \mathbf{x}(a) + \int_a^t K(t, \tau, \mathbf{x}(t), \mathbf{x}(\tau)) f_\theta(\tau, \mathbf{x}(\tau)) \mathrm{d}\tau,$$

Using the Lipschitz assumptions, we obtain

$$\|T[\mathbf{x}_1] - T[\mathbf{x}_2]\| \leq C(t - a)\|\mathbf{x}_1 - \mathbf{x}_2\|.$$

Choosing $\epsilon > 0$ small enough that $C\epsilon < 1$ and applying the Banach fixed-point theorem [83] yields a unique solution.

**Stability.** From (Eq. (15).), using the Lipschitz assumptions again as well as boundedness of $K$, it is readily seen that

$$\|\mathbf{x}(t) - \tilde{\mathbf{x}}(t)\|_2 \leq \|\mathbf{x}(a) - \tilde{\mathbf{x}}(a)\|_2 + C\int_a^t \|\mathbf{x}(\tau) - \tilde{\mathbf{x}}(\tau)\|_2 d\tau,$$

which gives the desired result by classical Grönwall inequality [84, Lemma B.9].

As in singular case, one can also show that the basic iteration method works well for the nonsingular case, that is, it is convergent with first order.

## E  FADE Examples

Here shows multiple FADE variants based on graph learning tasks. Inspired by the models in [19], we develop two variants, including Kat-GRAND and Kat-CDE. Similar to [8], Kat-GRAND includes two versions. One is Kat-GRAND-nl:

$$\int_a^t K(t, \tau, \mathbf{Y}(t))\Big((\mathbf{A}(\mathbf{Y}(t)) - \mathbf{I})\mathbf{Y}(t)\Big)\,\mathrm{d}\tau = \mathbf{Y}(t), \tag{22}$$

where $\mathbf{A}(\mathbf{Y}(t)) = (a_{i,j}(t))$ is given by a nonlinear attention mechanism. The other version is Kat-GRAND-l:

$$\int_a^t K(t, \tau, \mathbf{Y}(t))\,(-\mathbf{L}\mathbf{Y}(t))\,\mathrm{d}\tau = \mathbf{Y}(t), \tag{23}$$

where $\mathbf{L}$ is a time-invariant matrix, which is a linear FDE.

Furthermore, based on the CDE model [10], the Kat-CDE model has the following expression:

$$\int_a^t K(t, \tau, \mathbf{Y}(t))\Big(\mathbf{A}(\mathbf{Y}(t)) - \mathbf{I})\mathbf{Y}(t) + \mathrm{div}(\mathbf{V}(t) \circ \mathbf{Y}(t))\Big)\,\mathrm{d}\tau = \mathbf{Y}(t), \tag{24}$$

where the divergence operator $\mathrm{div}(\cdot)$ is introduced by [85], and $\circ$ stands for the element-wise product, also known as the Hadamard product. This model is crafted to handle heterophilic graphs, where connected nodes typically belong to different classes or exhibit distinct features.

## F  Graph Differential Equation Models

To better understand baseline models, this section primarily introduces several dynamic comparison networks based on graph learning tasks, namely GRAND [56], CDE [10], FROND [19], DRAGON [38] and NvoFDE [33].

**GRAND** [56]: The Graph Neural Diffusion (GRAND) model is a graph neural network framework inspired by the heat diffusion process, where information spreads across graph nodes similarly to how heat diffuses through a medium. Its governing differential equation is given by:

$$\frac{\mathrm{d}\mathbf{X}(t)}{\mathrm{d}t} = (\mathbf{A}(\mathbf{X}(t)) - \mathbf{I})\mathbf{X}(t), \tag{25}$$

where $\mathbf{A}(\mathbf{X}(t))$ is a learnable attention-based adjacency matrix, and $\mathbf{I}$ is the identity matrix. There are two variants. The update in (25) defines the **GRAND-nl** model, where the adjacency matrix $\mathbf{A}(\mathbf{X}(t))$ is nonlinear. Let $d_i = \sum_{j=1}^n W_{ij}$ and define the diagonal matrix $\mathbf{D}$ with $D_{ii} = d_i$. The random walk Laplacian is $\mathbf{L} = \mathbf{I} - \mathbf{W}\mathbf{D}^{-1}$. Thus in the simple case, we have the following **GRAND-l** model:

$$\frac{\mathrm{d}\mathbf{X}(t)}{\mathrm{d}t} = (\mathbf{W}\mathbf{D}^{-1} - \mathbf{I})\mathbf{X}(t) = -\mathbf{L}\mathbf{X}(t). \tag{26}$$

**CDE** [10]: In heterophilic graph, nodes often have diverse features, posing significant challenges for graph information processing. To address this issue, the authors introduced the convection-fiffusion equations (CDE) into GNNs,and then proposed the Neural CDE model. This model adaptively regulates the rate of information propagation between nodes, enabling selective information sharing among dissimilar neighbors. The corresponding mathematical formulation is given by:

$$\frac{\mathrm{d}\mathbf{X}(t)}{\mathrm{d}t} = (\mathbf{A}(\mathbf{X}(t)) - \mathbf{I})\mathbf{X}(t) + \mathrm{div}(\mathbf{V}(t) \circ \mathbf{X}(t)), \tag{27}$$

where $\mathbf{V}(t)$ denotes the velocity field, $\circ$ indicates the element-wise product, and $\mathrm{div}(\cdot)$ represents the divergence operator.

**FROND** [19]: The FROND framework extends traditional integer-order graph neural differential equations to fractional-order dynamics using the Caputo derivative:

$$D_t^\alpha \mathbf{X}(t) = \mathcal{F}(\mathbf{W}, \mathbf{X}(t)), \quad \alpha > 0, \tag{28}$$

where $\mathcal{F}$ defines the graph dynamics. By leveraging the non-local nature of fractional calculus, FROND captures long-range dependencies in node features. Similar to (25), (26) and (27), FROND has the following corresponding variants:

(1) **F-GROND-nl**

$$D_t^\alpha \mathbf{X}(t) = (\mathbf{A}(\mathbf{X}(t)) - \mathbf{I})\mathbf{X}(t), \quad 0 < \alpha \le 1. \tag{29}$$

(2) **F-GROND-l**

$$D_t^\alpha \mathbf{X}(t) = -\mathbf{L}\mathbf{X}(t), \quad 0 < \alpha \le 1. \tag{30}$$

(3) **F-CDE**

$$D_t^\alpha \mathbf{X}(t) = (\mathbf{A}(\mathbf{X}(t)) - \mathbf{I})\mathbf{X}(t) + \mathrm{div}(\mathbf{V}(t) \circ \mathbf{X}(t)), \quad 0 < \alpha \le 1. \tag{31}$$

**DRAGON** [38]: Unlike conventional continuous GNNs that rely on fixed integer or single fractional-order derivatives, DRAGON adopts a learnable distribution over derivative orders:

$$\int_a^b D^\alpha \mathbf{X}(t)\, d\mu(\alpha) = \mathcal{F}(\mathbf{W}, \mathbf{X}(t)), \tag{32}$$

where $[a, b]$ defines the domain of $\alpha$, $\mu$ is a learnable distribution, and $\mathcal{F}$ denotes the graph dynamics. Similar to (25), (26) and (27), DRAGON has the following corresponding variants:

(1) **D-GRAND-nl**

$$\int_0^1 D^\alpha \mathbf{X}(t)\, d\mu(\alpha) = (\mathbf{A}(\mathbf{X}(t)) - \mathbf{I})\mathbf{X}(t). \tag{33}$$

(2) **D-GRAND-l**

$$\int_0^1 D^\alpha \mathbf{X}(t)\, d\mu(\alpha) = \mathbf{L}\mathbf{X}(t). \tag{34}$$

(3) **D-CDE**

$$\int_0^1 D^\alpha \mathbf{X}(t)\, d\mu(\alpha) = (\mathbf{A}(\mathbf{X}(t)) - \mathbf{I})\mathbf{X}(t) + \mathrm{div}(\mathbf{V}(t) \circ \mathbf{X}(t)). \tag{35}$$

**NvoFDE** [33]: NvoFDE extends neural differential equation models by introducing a learnable variable-order derivative $\alpha(t, x(t))$ that dynamically adapts over time and feature space.

$$D_t^{\alpha(t, \mathbf{x}(t))} \mathbf{X}(t) = \mathcal{F}(\mathbf{W}, \mathbf{X}(t)), \quad 0 < \alpha(t, \mathbf{X}(t)) \le 1. \tag{36}$$

where $\mathcal{F}$ defines the graph dynamics. Similar to the above, there exist the following variants:

(1) **Nvo-GROND-nl**

$$D_t^{\alpha(t,\mathbf{x}(t))}\mathbf{X}(t) = (\mathbf{A}(\mathbf{X}(t)) - \mathbf{I})\mathbf{X}(t), \quad 0 < \alpha(t,\mathbf{X}(t)) \leq 1. \tag{37}$$

(2) **Nvo-GROND-l**

$$D_t^{\alpha(t,\mathbf{x}(t))}\mathbf{X}(t) = -\mathbf{L}\mathbf{X}(t), \quad 0 < \alpha(t,\mathbf{X}(t)) \leq 1. \tag{38}$$

(3) **Nvo-CDE**

$$D_t^{\alpha(t,\mathbf{x}(t))}\mathbf{X}(t) = (\mathbf{A}(\mathbf{X}(t)) - \mathbf{I})\mathbf{X}(t) + \mathrm{div}(\mathbf{V}(t) \circ \mathbf{X}(t)), \quad 0 < \alpha(t,\mathbf{X}(t)) \leq 1. \tag{39}$$

# G  Datasets and More Experiments on Graph for FADE Model

## G.1  Datasets and Setting

The datasets used in this paper are provided separately in Table 4 and Table 5.

Table 4: Dataset statistics used in Table 1 of the main text

| Dataset | Type | Classes | Features | Nodes | Edges |
|---|---|---|---|---|---|
| Cora | citation | 7 | 1433 | 2485 | 5069 |
| Citeseer | citation | 6 | 3703 | 2120 | 3679 |
| PubMed | citation | 3 | 500 | 19717 | 44324 |
| Coauthor CS | co-author | 15 | 6805 | 18333 | 81894 |
| Computers | co-purchase | 10 | 767 | 13381 | 245778 |
| Photo | co-purchase | 8 | 745 | 7487 | 119043 |
| CoauthorPhy | co-author | 5 | 8415 | 34493 | 247962 |
| OGB-Arxiv | citation | 40 | 128 | 169343 | 1166243 |
| Airport | tree-like | 4 | 4 | 3188 | 3188 |

Table 5: Dataset statistics used in Table 6

| Dataset | Nodes | Edges | Classes | Node Features |
|---|---|---|---|---|
| Roman-empire | 22662 | 32927 | 18 | 300 |
| Wiki-cooc | 10000 | 2243042 | 5 | 100 |
| Minesweeper | 10000 | 39402 | 2 | 7 |
| Questions | 48921 | 153540 | 2 | 301 |
| Workers | 11758 | 519000 | 2 | 10 |
| Amaon-ratings | 24492 | 93050 | 5 | 300 |

The authors revealed critical limitations in the commonly used benchmark datasets for evaluating models on heterophilic graphs [86]. To address this, they introduced several new datasets, such as Roman-empire, Wiki-cooc, Questions, Workers and Amazon-ratings. These datasets, sourced from different fields, have low homophily scores and display a variety of structural properties. We follow the experimental setup specified in the CDE model [10]. For the Workers, and Questions datasets,we employ the ROC-AUC score as the evaluation metric, as these tasks involve binary classification. Using the Amazon-ratings dataset as an example to reveal the experimental parameter settings, we set time = 3, step size = 1, learning rate = 0.001, weight decay = 0.0005, epoch = 1000 and dim = 128. The performance of our FADE-CDE model, is evaluated against several well-known baselines, including TDE-GNN[87], GRAND [56], GraphBel [85], NSD [88], ACMP[89], CDE [10], F-CDE [19], D-CDE[38] and Nvo-CDE[33].

## G.2  Node Classification on Heterophilic Graph

**Performance and Analysis**: In Table 6, we present experimental results on heterophilic graph datasets. It is evident that our model FADE-CDE achieves competitive or better performance, demonstrating its effectiveness. On Workers and Amazon-ratings datasets, FADE-CDE achieves the best performance among all compared models, outperforming the Nvo-CDE model by approximately 0.7% on both datasets. This advantage stems from the framework's flexibility in kernel function design, enabling it to capture more complex feature-updating dynamics.

Table 6: Node classification results(%). The best and the second-best result for each criterion are highlighted in red and blue, respectively.

| Model | Roman-empire | Wiki-cooc | Questions | Workers | Amazon-ratings |
|---|---|---|---|---|---|
| TDE-GNN[87] | 64.29±0.58 | 84.95±0.78 | 68.94±1.69 | 75.13±0.81 | 40.33±1.37 |
| GRAND-l[56] | 69.24±0.53 | 91.58±0.37 | 68.54±1.07 | 75.59±0.86 | 48.99±0.35 |
| GRAND-nl[56] | 71.60±0.58 | 92.03±0.46 | 70.67±1.28 | 75.33±0.84 | 45.05±0.65 |
| GraphBel[85] | 69.47±0.37 | 90.30±0.50 | 70.79±0.99 | 73.02±0.92 | 43.63±0.42 |
| NSD[88] | 77.50±0.67 | 92.06±0.40 | 69.25±1.15 | 79.81±0.99 | 37.96±0.20 |
| ACMP[89] | 71.27±0.59 | 92.68±0.37 | 71.18±1.03 | 75.03±0.92 | 44.76±0.52 |
| CDE[10] | 91.64±0.28 | 97.99±0.38 | **75.17±0.99** | 80.70±1.04 | 47.63±0.43 |
| F-CDE[19] | 93.06±0.55 | 98.73±0.68 | **75.17±0.99** | 82.68±0.86 | 49.01±0.56 |
| D-CDE[38] | **93.87±0.41** | 98.58±0.12 | **75.53±0.98** | 83.02±0.86 | 49.43±1.26 |
| Nvo-CDE[33] | 93.42±0.22 | **99.32±0.28** | 74.87±0.23 | **83.33±0.65** | **50.09±0.40** |
| FADE-CDE (ours) | **93.46±0.32** | **98.94±0.12** | 75.10±0.11 | **84.02±0.38** | **50.75±0.45** |

# H  Datasets and More Experiments for Traffic Forecasting

## H.1  Datasets and Setting

We evaluate the effectiveness of STDDE using five real-world traffic datasets: PeMSD7(M), PeMSD7(L), PeMS04, PeMS07, and PeMS08. These datasets are sourced from the Caltrans Performance Measurement System [59], which collects traffic flow data every 30 seconds. For analysis, the data is aggregated into 5-minute intervals, resulting in 288 time steps one day. A summary of the dataset statistics can be found in Table 7. These datasets are pre-divided into training, validation, and testing sets using a 6:2:2 ratio. The training procedure and hyperparameter settings are kept consistent with those reported in [4]. For instance, the model is trained for 200 epochs using the Adam optimizer.

Table 7: Datasets for Trafffic Forecasting

| Datasets | Sensors | Edges | Time Steps |
|---|---|---|---|
| PeMS04 | 307 | 340 | 16992 |
| PeMS07 | 883 | 866 | 28224 |
| PeMS08 | 170 | 295 | 17856 |
| PeMS07(M) | 228 | 1132 | 12672 |
| PeMS07(L) | 1026 | 10150 | 12672 |

Table 8: Forecasting error on PeMSD7

| Model | PeMSD7 | | |
|---|---|---|---|
| | MAE | RMSE | MAPE |
| HA[60] | 45.12 | 65.64 | 24.51% |
| VAR[60] | 50.22 | 75.63 | 32.22% |
| TCN[61] | 32.72 | 42.23 | 14.26% |
| DSANet[62] | 31.36 | 49.11 | 14.43% |
| AGCRN[63] | 22.37 | 36.55 | 9.12% |
| STFGNN[64] | 23.46 | 36.60 | 9.21% |
| Z-GCNETs[65] | 21.77 | 35.17 | 9.25% |
| STGODE[6] | 22.59 | 37.54 | 10.14% |
| STG-NCDE[4] | **20.53** | **33.84** | **8.80**% |
| STG-FADE (ours) | **20.46** | **33.70** | **8.94**% |

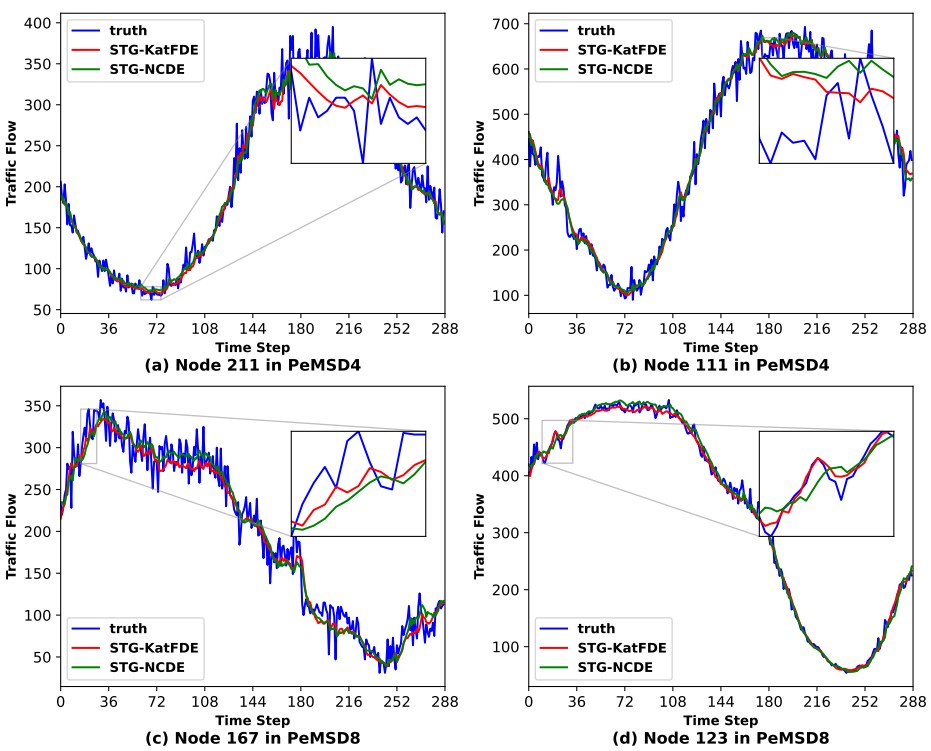

Figure 1: Traffic forecasting visualization in PeMSD4 and PeMSD8

## H.2 Experimental Results

From Table 8, our model STG-FADE achieves generally the best performance on PeMSD7 dataset. Compared to NDEs-based models, it demonstrates a stronger ability to capture complex dynamics. Figure 1 illustrates the predicted traffic flow from STG-FADE in comparison with STG-NCDE and the ground truth on PeMSD4 and PeMSD8 datasets. The horizontal axis denotes the time steps (5-minute intervals), and the vertical axis represents the traffic flow. A total of 288 time steps are selected, covering an entire 24-hour period.

Each subfigure corresponds to a specific node and is annotated with a zoomed-in region to highlight prediction differences in more dynamic or complex traffic periods. Overall, both models demonstrate a strong ability to follow the ground truth trends. However, the proposed STG-FADE consistently achieves closer alignment with the ground truth, especially in rapidly changing regions.

Particularly in Figure 1:

- Node 211 and Node 111 in PeMSD4 (top row) show that STG-FADE better captures sudden increases and local peaks, maintaining smoother yet accurate transitions.
- Node 167 and Node 123 in PeMSD8 (bottom row) further validate STG-FADE superiority, with visibly reduced error margins in congested and fluctuating segments, as shown in the zoom-in windows.

These results support the quantitative findings discussed in the main text and demonstrate the robustness and generalization capacity of STG-FADE across different traffic environments.

## H.3 Parameter Analysis

**Hidden Dimension Analysis**: Figure 2 presents the performance of STG-FADE on the PeMSD8 dataset with varying input feature dimensions: 16, 32, 64, and 128. It can be observed that as the feature dimension increases, the model's performance improves consistently across all three metrics. In particular, the lowest RMSE and MAE are achieved at dimension 128, indicating that

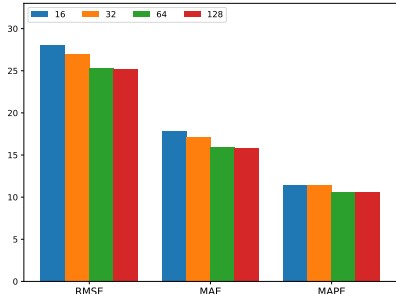

Figure 2: Performance comparison of STG-FADE with varying feature dimensions on the PeMSD8 dataset.

Table 9: Forecasting error metrics (MAE, RMSE, MAPE) for different step sizes on PeMSD4 and PeMSD8 datasets.

| Step size | PeMSD4 | | | PeMSD8 | | |
|---|---|---|---|---|---|---|
| | MAE | RMSE | MAPE | MAE | RMSE | MAPE |
| 0.5 | 19.48 | 31.34 | 12.94% | 17.45 | 27.55 | 11.07% |
| 1.0 | 19.88 | 31.83 | 13.14% | 17.10 | 26.83 | 10.66% |

higher-dimensional representations help capture more complex spatiotemporal patterns in traffic data. Notably, the improvement becomes more pronounced when increasing the dimension from 32 to 64, and then stabilizes between 64 and 128. These findings suggest that while increasing feature dimensionality benefits performance, the marginal gain diminishes beyond a certain point.

**Step Size Analysis**: Table 9 reports the forecasting error metrics of STG-FADE on the PeMSD4 and PeMSD8 datasets, evaluated with varying step sizes. As the step size increases, MAE increases from 19.48 to 19.88, and RMSE from 31.34 to 31.83 on PeMSD4; while on PeMSD8, MAE changes marginally from 17.45 to 17.10 and RMSE slightly decreases from 27.55 to 26.83. The experiments suggest that a moderately larger step size contributes to improved performance on PeMSD8, while it has the opposite effect on PeMSD4.

# I   Experiments for Urban Population Mobility and Biological Neural Spike Train Dynamics

## I.1   Urban Population Mobility Prediction

Post-disaster urban mobility dynamics exhibit complex patterns driven by both the disruptive disaster context and underlying habitual mobility. The original population mobility data from SafeGraph records daily movements between Census Block Groups (CBGs) during the period from August 1 to September 10, 2019. The authors aggregated these inter-CBG flows at the county level to obtain daily within-county and between-county population flows, representing intra-regional and inter-regional population flows on nodes and edges, respectively [90]. Based on this, we evaluated FADE on Florida's population mobility data during Hurricane Dorian. This dataset records daily inter-regional movements within Florida.

We compared FADE against baselines including LSTM [91], AGCRN [63], NDCN [92], CG-ODE [93], STG-NCDE [4], PatchTST [94] and CDGON [90]. Following [90], we utilize multiple evaluation metrics, including Mean Absolute Error (MAE), Normalized Root-Mean-Square Error (NRMSE), and Coefficient of Determination ($R^2$). As shown in Table 10, FADE achieves state-of-the-art performance. This is a quintessential use case for FADE: the fractional operator enables long memory, while the adaptive attention kernel allows dynamic weighing of recent disaster-related vs. routine historical patterns. The model can down-weight pre-disaster commuting patterns during the hurricane and re-integrate them during recovery.

## I.2   Biological Neural Spike Train Dynamics

We next tested FADE on biological time-series data of neural spike trains from multiple animals and brain regions. Neural systems are inherently history-dependent: the evolution of a neuron's membrane potential is influenced by prior inputs and spike events, and downstream firing patterns are shaped by this accumulated temporal context. We tested the experimental results on spiking datasets

Table 10: Average prediction error of different models on post-disaster urban mobility

| Model | MAE | NRMSE | $R^2$ |
|---|---|---|---|
| LSTM | 302884.0938 | 0.6689 | 0.5526 |
| AGCRN | 273478.9062 | 0.5377 | 0.7109 |
| NDCN | 406064.2812 | 0.3987 | 0.8411 |
| CG-ODE | 224787.1250 | 0.2936 | 0.9138 |
| STG-NCDE | 226016.1562 | 0.6683 | 0.5533 |
| PatchTST | 96736.0547 | 0.1632 | 0.9734 |
| CDGON | 59767.4805 | 0.0724 | 0.9948 |
| FADE (ours) | **24040.9355** | **0.0506** | **0.9974** |

Allen and Retina [95]. The Allen dataset contains spike train data from various brain regions and is designed to evaluate models for temporal and spatiotemporal neural activity classification. The Retina dataset provides spike trains from salamander retinal ganglion cells under four visual stimuli for stimulus-type classification. We compared FADE against baseline models, primarily including LSTM and Neural ODE [2]. As Table 11 shows, FADE outperforms standard recurrent and neural ODE models, highlighting its advantage in modeling nonlinear and temporally dependent neural systems.

Table 11: Test Accuracy of different models on neuron spike train classification %

| Dataset | LSTM | Neural ODE | FADE (ours) |
|---|---|---|---|
| Allen | 85.05 | 85.05 | **86.03** |
| Retina | 90.36 | 92.25 | **94.79** |

## J   Limitations and Broader Impact

Our key contribution is using a flexible kernel-attention fractional neural ODE that replaces fixed power-law memory kernels in traditional fractional differential equations. This framework FADE effectively unifies and extends integer-order, variable-order, and $\psi$-Caputo dynamics within a single continuous-depth framework. At the same time, we have validated its effectiveness and applicability across various real-world domains. However, there are some limitations of FADE. First, FADE is based on a deterministic equation. In practice, we often need to handle stochastic data and outputs with confidence intervals, which the current framework does not support. Second, the spatial interaction (i.e., the spatial partial differential components) at a given time is not explicitly included in the framework with direct incorporation of physical laws. To make FADE more generally applicable, future work should explicitly include spatial interactions and consider stochastic models instead of a purely deterministic formulation. From a broader societal perspective, FADE also carries potential risks of misuse or unintended consequences, such as in the transportation field. Therefore, it is essential to ensure that technological advancements lead to positive outcomes while keeping the application of FADE aligned with social values and ethical standards.

