# OpenReview forum: "Neural Fractional Attention Differential Equations"
_NeurIPS.cc/2025/Conference — NeurIPS 2025 poster_

### Official Review · Reviewer_jCMz · 2025-06-24

**Clarity:** 3
**Significance:** 3
**Originality:** 4
**Rating:** 4
**Confidence:** 4

**Summary:**

This paper proposes KatFDE, a generalized fractional neural ODE framework with learnable attention kernels. Instead of using fixed power-law kernels like in traditional fractional calculus, KatFDE introduces an attention-based kernel that adaptively weighs past hidden states. The framework unifies existing integer-, fractional-, and variable-order models, and provides theoretical guarantees (boundedness, convergence, etc.). Experiments on graph classification and traffic forecasting show improved or comparable performance.

**Questions:**

1) Can you show how far KatFDE actually looks into the past especially in time seires benchmarks? (e.g., attention heatmaps or span stats)
2) How does KatFDE compare with Informer/FEDformer/ContiFormer?
3) Ablation: what happens if you remove attention and just use fractional kernels?
4) Any insights from the learned kernels? Are they interpretable?

**Ethical Concerns:**

["NO or VERY MINOR ethics concerns only"]

**Final Justification:**

The authors' rebuttal, which includes new, well-designed experiments on challenging time-series tasks, has thoroughly addressed my primary concern regarding the narrow scope of the initial evaluation. The results from these new benchmarks provide compelling evidence for the KatFDE framework's superior modeling capacity and generality. However, the paper's overall contribution, while significantly improved, does not entirely warrant a change in my initial assessment of its impact and scope. Therefore, I have decided to maintain my original score.

**Limitations:**

Limited experiment scope and lack of behavioral analysis. No strong evidence that long-range dependencies are captured better than existing methods. Claims about generality aren’t fully supported by the benchmarks used.

**Paper Formatting Concerns:**

While there is no major formatting issue, the conclusion section is quite brief and lacks the depth expected (e.g., reflection on implications or limitations). The presentation of Theorem and Lemma environments could be improved by clearer formatting or boxed environments to improve readability.

**Quality:**

3

**Strengths And Weaknesses:**

+ Conceptually clean integration of fractional dynamics and attention.
+ Theoretical analysis is solid. Unified view is elegant.
+ Better performance than previous FDE models like FROND and NvoFDE.

– Claims about capturing long-range dependencies aren’t backed by empirical evidence (no attention weight analysis or memory span plots).
– Evaluation is narrow: just two tasks. No irregular or noisy time series.
– No comparison with recent Transformer-based time-series models (Informer, FEDformer, etc.).
– Little discussion about model behavior or efficiency.
– Conclusion is underwhelming.

---

> ### Author Rebuttal · Authors · 2025-07-31
>
> ## I. New Supporting Experimental Results
>
> We appreciate the reviewer’s critique of the experimental evaluation and thank them for motivating us to test our framework on more complex, non-linear, temporally evolving dynamical systems. The marginal advantage reported in the original paper may be attributable to the evaluated system not being sufficiently challenging to fully demonstrate the strengths of our framework.
>
> To further demonstrate KatFDE’s generality and capability, we have added three additional experiments on complex temporally evolving systems.
>
> __1. Fluid Flow Prediction:__
>
> We evaluated KatFDE on turbulent boundary-layer flow [H1], with velocity fields measured using particle image velocimetry at five Reynolds numbers (Re = 600, 980, 1370, 1780, 2220), each containing approximately 6,000 snapshots. We trained on four Reynolds numbers and tested on the fifth. Models observed 6 consecutive snapshots to predict the next 6, using a CNN encoder–decoder for spatial feature extraction and LSTM, Transformer, Neural ODE, or our KatFDE for latent state temporal evolution.
> Turbulent flows exhibit strong memory effects due to the cascade of energy across different scales and the persistence of coherent structures. The non-local temporal dependencies in turbulence make it an ideal testbed for our attention-based fractional framework, which can adaptively weight historical flow states based on their relevance to current dynamics. The preliminary prediction results are summarized in **Table R1**, demonstrating that KatFDE's adaptive memory mechanism effectively captures the nonlinear multi-scale temporal dependencies inherent in turbulent flows, outperforming other approaches.
>
> | Model       | LSTM+CNN | Transformer+CNN | ODE+CNN | Ours+CNN |
> |--------------|----------|-----------------|---------|----------|
> | RMSE     | 1.5035   | 0.4701           | 0.5962  | **0.3215** |
> | MAE      | 0.6027   | 0.3532           | 0.2799  | **0.2041** |
>
> **Table R1**: Average prediction error of different models on turbulent vector field prediction.
>
> [H1] Towne, A., Dawson, S., Brès, G. A., Lozano-Durán, A., Saxton-Fox, T., Parthasarthy, A., Biler, H., Jones, A. R., Yeh, C.-A., Patel, H., Taira, K. (2022). A database for reduced-complexity modeling of fluid flows. AIAA Journal 61(7): 2867-2892.
>
>
> __2. Urban Population Mobility Prediction:__
>
> Post-disaster urban mobility dynamics exhibit complex patterns driven by both the disruptive disaster context and underlying habitual mobility. We evaluated KatFDE on Florida’s population mobility data during Hurricane Dorian (August 1–September 10, 2019), sourced from SafeGraph and reported in [H2].
> This dataset records daily inter-regional movements within Florida. We compared KatFDE against baselines including LSTM, Transformer, and neural ODE variants. As shown in **Table R2**, KatFDE achieves state-of-the-art performance.
> This is a quintessential use case for KatFDE: the fractional operator enables long memory, while the adaptive attention kernel allows dynamic weighing of recent disaster-related vs. routine historical patterns. The model can down-weight pre-disaster commuting patterns during the hurricane and re-integrate them during recovery.
>
> | Model         | MAE         | NRMSE  | R²     |
> |--------------|-------------|--------|--------|
> | LSTM          | 302884.0938 | 0.6689 | 0.5526 |
> | AGCRN         | 273478.9062 | 0.5377 | 0.7109 |
> | NDCN          | 406064.2812 | 0.3987 | 0.8411 |
> | CG-ODE        | 224787.1250 | 0.2936 | 0.9138 |
> | STG-NCDE      | 226016.1562 | 0.6683 | 0.5533 |
> | PatchTST      |  96736.0547 | 0.1632 | 0.9734 |
> | CDGON         |  59767.4805 | 0.0724 | 0.9948 |
> | **KatFDE (ours)** | **24040.9355** | **0.0506** | **0.9974** |
>
> **Table R2**: Average prediction error of different models on post-disaster urban mobility.
>
> [H2] Li, J., etc. Physics-informed neural ode for post-disaster mobility recovery. In KDD 2024.
>
> __3. Biological Neural Spike Train Dynamics:__
>
> We next tested KatFDE on biological time-series data of neural spike trains from multiple animals and brain regions. Neural systems are inherently history-dependent: the evolution of a neuron's membrane potential is influenced by prior inputs and spike events, and downstream firing patterns are shaped by this accumulated temporal context.
> We tested the experimental results on spiking datasets Allen and Retina [H3]. The Allen dataset contains spike train data from various brain regions and is designed to evaluate models for temporal and spatiotemporal neural activity classification. The Retina dataset provides spike trains from salamander retinal ganglion cells under four visual stimuli for stimulus-type classification.  As **Table R3** shows, KatFDE outperforms standard recurrent and neural ODE models, highlighting its advantage in modeling nonlinear and temporally dependent neural systems.
>
>
>
> | Dataset | LSTM  | Neural ODE | KatFDE (ours)    |
> |--------|--------|------------------|---------|
> | Allen  | 85.05 | 85.05      | **86.03** |
> | Retina   | 90.36| 92.25    | **94.79** |
>
> **Table R3**: Test Accuracy of different models on neuron spike train classification (%)
>
>
> [H3] Lazarevich I, Prokin I, Gutkin B, et al. Spikebench: An open benchmark for spike train time-series classification[J]. PLOS Computational Biology, 2023, 19(1): e1010792.
>
> ***
>
> ## II. Ablation Study: If Remove Learnable Attention:
>
> We emphasize that the comparisons in Table 1 already provide a clear ablation analysis. This is fully consistent with our explanation of the novelty, which is framed relative to the first-order derivative, the constant-order Caputo derivative, and the variable-order derivative definitions. **Each of these progressively introduces greater advantages and flexibility to the continuous system. Our framework fully generalizes and unifies all of these approaches.**
>
> The ablation study is shown in **Table R4**, reproduced from Table 1 in the paper.
>
> | Model         | Comparison                | Cora         | Citeseer     | Computer     |
> |---------------|---------------------------|--------------|--------------|--------------|
> | GRAND-l       | first-order ODE [48]      | 83.6±1.0     | 73.4±0.5     | 83.7±1.2     |
> | F-GRAND-l     | constant-order FDE [18]   | 84.8±1.1     | 74.0±1.5     | 84.4±1.5     |
> | D-GRAND-l     | constant-order FDE [51]   | 85.1±1.3     | 74.5±1.1     | 87.3±1.3     |
> | Nvo-GRAND-l   | variable-order FDE [32]   | 86.0±0.5     | 75.6±0.8     | 87.9±0.8     |
> | **KatFDE-l**      | **our unified learnable FDE** | **86.4±0.5** | **76.1±0.6** | **88.3±0.9** |
>
>
> | Model         | Comparison                | Cora         | Citeseer     | Computer     |
> |---------------|---------------------------|--------------|--------------|--------------|
> | GRAND-nl      | first-order ODE [48]      | 82.3±1.6     | 70.9±1.0     | 82.4±2.1     |
> | F-GRAND-nl    | constant-order FDE [18]   | 83.2±1.1     | 74.7±1.9     | 84.1±0.9     |
> | D-GRAND-nl    | constant-order FDE [51]   | 83.9±1.3     | 74.8±1.6     | 87.1±1.0     |
> | Nvo-GRAND-nl  | variable-order FDE [32]   | 85.4±1.0     | 75.9±0.6     | 87.2±1.4     |
> | **KatFDE-nl**     | **our unified learnable FDE** | **86.0±0.4** | **76.2±0.8** | **87.7±0.9** |
>
> **Table R4**: Ablation study among differential-equation-based continuous models. _The results show that through generalization, our model achieves an advantage._
>
> ***
>
> ## III. Interpretability of the Kernel
>
> The complex fully learnable kernel is inherently difficult to interpret. We will include attention heatmaps in the revised version to provide better visualization.
> Several traditional fractional derivative operators (see Eq10. and Eq12. in the paper) define
> $$
> K(t, \tau, \mathbf{x}(t), \mathbf{x}(\tau))=\frac{\psi^{\prime}(\tau)(\psi(t)-\psi(\tau))^{\alpha-1}}{\Gamma(\alpha)} \quad \text { or } \quad K(t, \tau, \mathbf{x}(t), \mathbf{x}(\tau)) = \frac{(t-\tau)^{\alpha-1}}{\Gamma(\alpha)},
> $$
> which depend only on the temporal distance between states. In these cases, __closer states exert a stronger influence on each other than more distant ones__. For a general kernel $K$, however, direct interpretability is considerably more challenging.
>
> ***
>
> ## IV. Comparison with Informer/FEDformer/ContiFormer
>
> As demonstrated before, this work aims to propose an innovative framework that is flexible and scalable for a wide range of applications. Within this framework, we verify the performance of the fusion of FDE model and scaled dot product attention mechanism. Informer[w1], FEDformer[w2] and ContiFormer[w3] are proposed to improve transformer either by reducing the complexity from $O(L^2)$ to $O(L\log L)$ or even $O(L)$ with ProbSparse self-attention or frequency enhanced attention with Fourier/wavelet transform respectively, or using averaged continuous inner product instead of dot product in transformer to tackle irregular time series. These advanced strategies are essentially compatible with our framework. Specifically, one can readily use attention mechanism proposed in Informer, FEDformer and ContiFormer by specifying corresponding kernels in (20). Therefore, we will work with Informer,FEDformer and ContiFormer in the current framework, instead of comparing with them.
>
> [w1]Zhou Haoyi, et al. Informer: Beyond Efficient Transformer for Long Sequence Time-Series Forecasting, AAAI2021.
>
> [w2]Zhou tian, et al. FEDformer: Frequency Enhanced Decomposed Transformer for Long-term Series Forecasting, ICML2022.
>
> [w3] Chen Yuqi,et al. ContiFormer: Continuous-Time Transformer for Irregular Time Series Modeling, NeurlPS2023.

---

> > ### Comment · Reviewer_jCMz · 2025-08-09
> >
> > I thank the authors for their detailed rebuttal. The new experiments on fluid flow, urban mobility, and biological neural spike trains are a valuable addition to the paper. They effectively demonstrate the generality and capability of the KatFDE framework on more complex, temporally evolving systems, which was a key weakness in my initial review. The new results show a substantial performance advantage over baselines, providing strong empirical evidence for the model's ability to handle complex, history-dependent dynamics.

---

> > > ### Author Response · Authors · 2025-08-09
> > > **Thank you for your support**
> > >
> > > Dear Reviewer jCMz,
> > >
> > > Thank you for your thorough review and invaluable feedback. We are glad that our responses have addressed your concerns. Your insightful suggestions have significantly improved the quality of our paper, and we sincerely appreciate your guidance.
> > >
> > > Sincerely,
> > >
> > > The Authors of Paper 19675

---

### Official Review · Reviewer_jEHE · 2025-07-13

**Clarity:** 2
**Significance:** 2
**Originality:** 3
**Rating:** 4
**Confidence:** 3

**Summary:**

This paper presents KatFDE, a neural model that replaces fixed fractional kernels with a learnable attention kernel so it can decide which past information matters. It proves the solver is sound and reports higher accuracy than earlier integer-order and fixed-kernel fractional models on graph node-classification and traffic-forecasting benchmarks.

**Questions:**

What are the training and inference NFE counts? What is the wall-clock time for running KatFDE, along with the GPU model, batch size, and solver tolerances used? What is KatFDE’s peak memory consumption? Compared with vanilla Caputo solvers and other baselines, does KatFDE require fewer resources to run?

Compared to other non–differential equation methods (such as DDGCRN [1] and Hierarchical‑Attention‑LSTM [2]), what advantages does KatFDE offer for tasks like traffic forecasting?

[1]https://www.sciencedirect.com/science/article/pii/S0031320323003710?casa_token=vWU0ZxgyAdEAAAAA:fjEXyBsi65YNLn1MWuAUrFlvrtQouPXFEJPRKd_jzk1b5T_2qo7qgBezHmiowdNVXhodvpittQ

[2] https://arxiv.org/abs/2201.05760

**Ethical Concerns:**

["NO or VERY MINOR ethics concerns only"]

**Final Justification:**

The authors address most of my concerns by providing extensive experimental results on real-world tasks. KatFDE demonstrates impressive performance compared to numerous benchmarks. The inclusion of training details enhances the paper's clarity, and the promised visualizations will further improve readability. Although the time complexity is still missing, the authors report memory usage, which is comparable to Transformer + CNN models. Given the paper’s scope and KatFDE’s strong performance, I consider this a borderline accept.

**Limitations:**

Yes.

**Paper Formatting Concerns:**

No.

**Quality:**

2

**Strengths And Weaknesses:**

**Strength**:
The idea of using a kernel-attention fractional neural ODE that replaces fixed power-law memory kernels with learnable attention mechanisms is novel. This method effectively unifying and extending integer-, variable-order, and ψ-Caputo dynamics within a single continuous-depth framework. While I did not examine every proof in detail, the theoretical foundation of the paper appears robust and well-developed.

**Weakness**:
- There is a typo on line 157 (“andmaps”).

- Several important details are missing, such as the choice of step size $h$, the window size used in KatFDE, specifications of KatFDE-I and KatFDE-nl, training settings, and model architecture.

- The paper also lacks an analysis of computational complexity and memory usage. Since standard self-attention scales with $O(T^2)$ time and $O(T^2)$ memory, KatFDE’s use of it raises concerns about efficiency. The absence of such information is a major concern.

- Additionally, the absence of visualizations makes the paper harder to follow. Lastly, the discussion would benefit from including the following related works \[1–3].

[1] Zhao, Chunna, et al. "FLRNN-FGA: Fractional-Order Lipschitz Recurrent Neural Network with Frequency-Domain Gated Attention Mechanism for Time Series Forecasting." Fractal and Fractional 8.7 (2024): 433.

[2] Vellappandi, Madasamy, and Sangmoon Lee. "Physics-informed neural fractional differential equations." Applied Mathematical Modelling 145 (2025): 116127.

[3] Fernandez, Arran, Mehmet Ali Özarslan, and Dumitru Baleanu. "On fractional calculus with general analytic kernels." Applied Mathematics and Computation 354 (2019): 248-265.

---

> ### Author Rebuttal · Authors · 2025-07-31
>
> ## Experimental details
>
> Some experimental details are presented in the appendix, and more will be added for completeness. The codebase for reproducing our experiments with exact parameters will be fully public.
> For brevity, we describe the setup for the node classification task on graphs. The experiments are implemented using the PyTorch framework on a single NVIDIA RTX4090 24GB GPU. For different datasets, parameters such as time steps and evolution time are slightly adjusted. Taking the Cora dataset as an example, we set t = 25, step size = 1, learning rate = 0.01, weight decay = 0.05, epoch = 800, and dim = 256. Following [48, 18], the model architectures of KatFDE-l and KatFDE-nl are provided in Appendix F. The visualization results are shown in Appendix I. If the paper is accepted, we will move the visualization results into the main text. Furthermore, we will include attention heatmaps in the revised version to provide better visualization, as suggested by Reviewer jCMz.
>
> ***
>
>
> ## Computational complexity and memory usage.
>
> The main focus of this work is to propose a unified framework of FDE methods from a new perspective and to verify its effectiveness. This framework opens new possibilities, such as fusing the attention mechanism with the memory kernel of FDE, which is investigated in this work. We focus only on the standard attention mechanism. Therefore, the extra computational complexity and memory usage from the attention mechanism are inevitable and standard.
>
>
> To further demonstrate KatFDE’s generality and capability, we have added three additional experiments on complex temporally evolving systems (we refer the reviewer to the full results presented in the top rebuttal to Reviewer ikK2). Here we present the **fluid flow prediction task with accuracy and memory consumption comparison:**
>
> We evaluated KatFDE on turbulent boundary-layer flow [H1], with velocity fields measured using particle image velocimetry at five Reynolds numbers (Re = 600, 980, 1370, 1780, 2220), each containing approximately 6,000 snapshots. We trained on four Reynolds numbers and tested on the fifth. Models observed 6 consecutive snapshots to predict the next 6, using a CNN encoder–decoder for spatial feature extraction and LSTM, Transformer, Neural ODE, or our KatFDE for latent state temporal evolution.
> Turbulent flows exhibit strong memory effects due to the cascade of energy across different scales and the persistence of coherent structures. The non-local temporal dependencies in turbulence make it an ideal testbed for our attention-based fractional framework, which can adaptively weight historical flow states based on their relevance to current dynamics.
>
> The preliminary prediction results are summarized in **Table R5**, demonstrating that KatFDE's adaptive memory mechanism effectively captures the nonlinear multi-scale temporal dependencies inherent in turbulent flows, outperforming other approaches. The GPU memory usage of KatFDE is slightly higher than the baseline LSTM and ODE models, but comparable to the Transformer. This is expected since the extra computational complexity and memory usage from the attention mechanism are inevitable and standard. Importantly, with nearly the same GPU memory usage, our continuous KatFDE framework performs much better than the standard Transformer.
>
>
> To alleviate GPU memory usage, due to the flexibility of KatFDE, one can incorporate efficient attention mechanisms such as [4], which can reduce the computational cost from $O\left(L^2\right)$ to $O(L \log L)$ in practice. However, investigating this type of improvement and other variants is beyond the scope of this work and will be left for future research.
>
> | Model       | LSTM+CNN | Transformer+CNN | ODE+CNN | ``Our KatFDE+CNN`` |
> |--------------|----------|-----------------|---------|----------|
> | RMSE     | 1.5035   | 0.4701           | 0.5962  | **0.3215** |
> | MAE      | 0.6027   | 0.3532           | 0.2799  | **0.2041** |
> | Training Memory | 11372MiB   | 14142MiB          | 11250MiB  | 14722MiB |
>
> **Table R5**: Average prediction error and memory consumption of different models on turbulent vector field prediction.
>
> ***
>
>
> ## Advantages of KatFDE compared to the references
>
> Recall that KatFDE is built from a general integral equation that explicitly incorporates the attention mechanism [3], while DDGCRN [1] and Hierarchical‑Attention‑LSTM [2] are not equation-based and rely on LSTM architectures. The advantages of KatFDE for tasks like traffic forecasting and **the new tested spatio-temporally continuous non-linear temporally evolving systems** (we refer the reviewer to the full results presented in the top rebuttal to Reviewer ikK2) are threefold:
> i) The forecasting data may follow some first principles and thus lead to an equation. Our integral equation is general enough to cover a wide range of such principles, and the kernel is set to be learnable so that KatFDE can capture them accurately. In this context, KatFDE is expected to perform better;
> ii) KatFDE naturally inherits temporal relations from the integral equation and incorporates the attention mechanism for temporal modeling. Both are well suited for time series data with long-range dependencies, which are hard to model under LSTM frameworks;
> iii) KatFDE is more flexible and compatible with recent efficient attention mechanisms [4].
>
> [1]Weng Wenchao, et al. A decomposition dynamic graph convolutional recurrent network for traffic forecasting[J]. Pattern Recognition, 2023, 142: 109670.
>
> [2]Zhang Tianya. Network Level Spatial Temporal Traffic Forecasting with Hierarchical Attention LSTM (HierAttnLSTM)
>
> [3]A. Vaswani, et al. Attention is all you need, NeurlPS2017.
>
> [4]Zhou Haoyi, et al. Informer: Beyond Efficient Transformer for Long Sequence Time-Series Forecasting, AAAI2021.

---

> > ### Comment · Reviewer_jEHE · 2025-08-01
> >
> > Thank you for the informative rebuttal. Additional experimental results are impressive and thank you for providing experiment details. I will consider raise my score.

---

> > > ### Author Response · Authors · 2025-08-01
> > > **thank you for your support!**
> > >
> > > Dear Reviewer jEHE,
> > >
> > > Thank you for your invaluable support and the time you devoted to reviewing our manuscript! We sincerely appreciate your insightful suggestions, which have greatly enhanced the quality of our paper!
> > >
> > > Sincerely,
> > >
> > > The Authors of Paper 19675

---

### Official Review · Reviewer_Qt1E · 2025-07-13

**Clarity:** 1
**Significance:** 2
**Originality:** 3
**Rating:** 4
**Confidence:** 4

**Summary:**

This paper introduces KatFDE, a new framework for modeling continuous-time dynamics. The problem is that normal neural ODEs and also the fractional neural ODEs cannot well model systems with memory. The proposed method, KatFDE, uses a fractional differential equation but replaces the fixed mathematical kernel with a learnable attention kernel. This lets the model decide which past states are important for the current state. The authors give theory to show the model is well-posed and stable, and they test it on graph node classification and traffic forecasting, showing it can perform better than other methods.

**Questions:**

1. The paper claims the framework is general for systems with memory. Can you give concrete examples of systems (e.g., in physics or finance) where older FDE models do not work well but KatFDE would be better because of the adaptive kernel? Explaining this would make the motivation stronger. If you give good answer, I will raise my score.
2. The experimental results are not consistent. For example, on some datasets your model is not better than the baseline. Can you give an analysis why the model performs well on some datasets but not others? This is needed to understand where the model is useful. A good analysis would make me raise the score.
3. The ablation study is missing. This is a critical experiment to understand the value of the new attention kernel. Please, can you add a study that compares your proposed kernel to simpler versions? This would justify the model's complexity. If this is added, I would raise my score.
4. The "Related Work" section is in the appendix. This is not standard. Can you move this section into the main paper? This is a major issue with the structure. Fixing this is important and would make me reconsider my score.
5. In Section 3, you mix the review of known attention with your proposed model. To make it clear, can you change Section 3.1.1 to separate the explanation of standard attention from your new fractional attention kernel? This will help me to see better your contribution.

**Ethical Concerns:**

["NO or VERY MINOR ethics concerns only"]

**Final Justification:**

The authors have provided extensive experiment results on real-world datasets against a wide range of compelling baselines, and have made clarifications about the ablation studies. The authors are encouraged to update their manuscript to include related works and be specific about the novelty of their architecture. If possible, please move some of the mathematical details to the appendix for ease of reading.

**Limitations:**

yes

**Quality:**

2

**Strengths And Weaknesses:**

Strengths:
- Originality: The idea to use a learnable neural attention inside the fractional operator is a novel contribution. It has potential for modeling complex systems.
- Theoretical Support: The paper gives rigorous mathematical proofs for the KatFDE framework. This analysis of boundedness and well-posedness gives confidence in the method.
- Generalization: The framework is a good generalization that connects integer-order ODEs, fractional ODEs, and attention mechanisms in one formulation.

Weaknesses:
- Clarity: The paper is written in a style that is hard to read. It is like a math textbook, not a research paper. For example, putting the definition of a derivative (Equation 1) is not necessary. Also, the "Related Work" section is in the appendix, which is very unorthodox.
- Experiments: The experimental part has major flaws. I see no ablation study to show why the attention kernel is important. The improvement over other models is also not consistent across all datasets, which makes me question if the model is truly effective.
- Presentation of Novelty: It is hard to understand what is the exact new contribution. The paper mixes the explanation of the model with review of well-known things like scaled dot-product attention, so the novelty is not clear.

---

> ### Author Rebuttal · Authors · 2025-07-31
>
> ## I. Novelty, Clarity, and Writing Style
>
> 1. Our key contribution is using a flexible kernel-attention fractional neural ODE that replaces fixed power-law memory kernels in traditional fractional differential equations. This framework effectively **unifies and extends integer-order, variable-order, and  $\psi$-Caputo dynamics within a single continuous-depth framework.** Furthermore, we **provide rigorous theoretical foundations** for our framework, including boundedness under both singular and nonsingular cases. We establish the well-posedness of the resulting neural integral equations using Banach fixed-point arguments, ensuring solution uniqueness under appropriate conditions.
> The scaled dot-product attention mechanism itself is well known, and _we do not claim novelty in its formulation_, but showing its effectiveness as a learnable kernel in fractional derivatives is new and one of our contributions.
>
> 2. Writing Style:  Our paper aims to bridge the mathematics and machine learning communities, where some readers may be unfamiliar with fractional definitions. We therefore **introduce the concepts progressively**, starting with the simple first-order definition, then moving to the constant-order Caputo definition, and finally to the variable- and $\psi$-order definitions. We explicitly outline these derivatives because _their comparison naturally motivates our general KatFDE formulation_, as discussed prior to Eq. (15).
>
> 3. We will bring the related work into the main paper if it is accepted, since there is allowance for one more page. Thank you for your suggestion.
>
>
> ***
>
> ## II. New Supporting Systems with Experimental Results
>
> We appreciate the reviewer’s critique of the experimental evaluation and thank them for motivating us to test our framework on more complex, non-linear, temporally evolving dynamical systems. The marginal advantage reported in the paper draft may be attributable to the evaluated datasets not being sufficiently challenging to fully demonstrate the strengths of our framework.
>
> To further demonstrate KatFDE’s generality and capability, we have added three additional experiments on complex temporally evolving systems.
>
> __1. Fluid Flow Prediction:__
>
> We evaluated KatFDE on turbulent boundary-layer flow [H1], with velocity fields measured using particle image velocimetry at five Reynolds numbers (Re = 600, 980, 1370, 1780, 2220), each containing approximately 6,000 snapshots. We trained on four Reynolds numbers and tested on the fifth. Models observed 6 consecutive snapshots to predict the next 6, using a CNN encoder–decoder for spatial feature extraction and LSTM, Transformer, Neural ODE, or our KatFDE for latent state temporal evolution.
> Turbulent flows exhibit strong memory effects due to the cascade of energy across different scales and the persistence of coherent structures. The non-local temporal dependencies in turbulence make it an ideal testbed for our attention-based fractional framework, which can adaptively weight historical flow states based on their relevance to current dynamics. The preliminary prediction results are summarized in **Table R1**, demonstrating that KatFDE's adaptive memory mechanism effectively captures the nonlinear multi-scale temporal dependencies inherent in turbulent flows, outperforming other approaches.
>
> | Model       | LSTM+CNN | Transformer+CNN | ODE+CNN | ``Our KatFDE+CNN`` |
> |--------------|----------|-----------------|---------|----------|
> | RMSE     | 1.5035   | 0.4701           | 0.5962  | **0.3215** |
> | MAE      | 0.6027   | 0.3532           | 0.2799  | **0.2041** |
>
> **Table R1**: Average prediction error of different models on turbulent vector field prediction.
>
> [H1] Towne, A., Dawson, S., Brès, G. A., Lozano-Durán, A., Saxton-Fox, T., Parthasarthy, A., Biler, H., Jones, A. R., Yeh, C.-A., Patel, H., Taira, K. (2022). A database for reduced-complexity modeling of fluid flows. AIAA Journal 61(7): 2867-2892.
>
>
> __2. Urban Population Mobility Prediction:__
>
> Post-disaster urban mobility dynamics exhibit complex patterns driven by both the disruptive disaster context and underlying habitual mobility. We evaluated KatFDE on Florida’s population mobility data during Hurricane Dorian (August 1–September 10, 2019), sourced from SafeGraph and reported in [H2].
> This dataset records daily inter-regional movements within Florida. We compared KatFDE against baselines including LSTM, Transformer, and neural ODE variants. As shown in **Table R2**, KatFDE achieves state-of-the-art performance.
> This is a clear use case for KatFDE: the fractional operator enables long memory, while the adaptive attention kernel allows dynamic weighing of recent disaster-related vs. routine historical patterns.
>
> | Model         | MAE         | NRMSE  | R²     |
> |--------------|-------------|--------|--------|
> | LSTM          | 302884.0938 | 0.6689 | 0.5526 |
> | AGCRN         | 273478.9062 | 0.5377 | 0.7109 |
> | NDCN          | 406064.2812 | 0.3987 | 0.8411 |
> | CG-ODE        | 224787.1250 | 0.2936 | 0.9138 |
> | STG-NCDE      | 226016.1562 | 0.6683 | 0.5533 |
> | PatchTST      |  96736.0547 | 0.1632 | 0.9734 |
> | CDGON         |  59767.4805 | 0.0724 | 0.9948 |
> | **KatFDE (ours)** | **24040.9355** | **0.0506** | **0.9974** |
>
> **Table R1**: Average prediction error of different models on post-disaster urban mobility.
>
> [H2] Li, J., etc. Physics-informed neural ode for post-disaster mobility recovery. In KDD 2024.
>
> __3. Biological Neural Spike Train Dynamics:__
>
> We next tested KatFDE on biological time-series data of neural spike trains from multiple animals and brain regions. Neural systems are inherently history-dependent: the evolution of a neuron's membrane potential is influenced by prior inputs and spike events, and downstream firing patterns are shaped by this accumulated temporal context.
> We tested the experimental results on spiking datasets Allen and Retina [H3]. The Allen dataset contains spike train data from various brain regions and is designed to evaluate models for temporal and spatiotemporal neural activity classification. The Retina dataset provides spike trains from salamander retinal ganglion cells under four visual stimuli for stimulus-type classification. As **Table R3** shows, KatFDE outperforms standard recurrent and neural ODE models, highlighting its advantage in modeling nonlinear and temporally dependent neural systems. Due to limited time during the rebuttal period, we will include more baselines in the final version.
>
>
> | Dataset | LSTM  | Neural ODE | KatFDE (ours)    |
> |--------|--------|------------------|---------|
> | Allen  | 85.05 | 85.05      | **86.03** |
> | Retina    | 90.36| 92.25    | **94.79** |
>
> **Table R3**: Test Accuracy of different models on neuron spike train classification (%)
>
>
> [H3] Lazarevich I, Prokin I, Gutkin B, et al. Spikebench: An open benchmark for spike train time-series classification[J]. PLOS Computational Biology, 2023, 19(1): e1010792.
>
> ***
>
> ## III. Clarify Ablation Study:
>
> We emphasize that the baselines in Table 1 in the paper already provide a clear ablation analysis. This is fully consistent with our explanation of the novelty, which is framed relative to the first-order derivative, the constant-order Caputo derivative, and the variable-order derivative definitions. **Each of these progressively introduces greater advantages and flexibility to the continuous system. Our framework fully generalizes and unifies all of these approaches, and shows clear advantages.**
>
> The ablation study is shown in **Table R4**, reproduced from Table 1 in the paper.
>
> | Model         | Comparison                | Cora         | Citeseer     | Computer     |
> |---------------|---------------------------|--------------|--------------|--------------|
> | GRAND-l       | first-order ODE [48]      | 83.6±1.0     | 73.4±0.5     | 83.7±1.2     |
> | F-GRAND-l     | constant-order FDE [18]   | 84.8±1.1     | 74.0±1.5     | 84.4±1.5     |
> | D-GRAND-l     | constant-order FDE [51]   | 85.1±1.3     | 74.5±1.1     | 87.3±1.3     |
> | Nvo-GRAND-l   | variable-order FDE [32]   | 86.0±0.5     | 75.6±0.8     | 87.9±0.8     |
> | **KatFDE-l**      | **our unified learnable FDE** | **86.4±0.5** | **76.1±0.6** | **88.3±0.9** |
>
>
> | Model         | Comparison                | Cora         | Citeseer     | Computer     |
> |---------------|---------------------------|--------------|--------------|--------------|
> | GRAND-nl      | first-order ODE [48]      | 82.3±1.6     | 70.9±1.0     | 82.4±2.1     |
> | F-GRAND-nl    | constant-order FDE [18]   | 83.2±1.1     | 74.7±1.9     | 84.1±0.9     |
> | D-GRAND-nl    | constant-order FDE [51]   | 83.9±1.3     | 74.8±1.6     | 87.1±1.0     |
> | Nvo-GRAND-nl  | variable-order FDE [32]   | 85.4±1.0     | 75.9±0.6     | 87.2±1.4     |
> | **KatFDE-nl**     | **our unified learnable FDE** | **86.0±0.4** | **76.2±0.8** | **87.7±0.9** |
>
> **Table R4**: Ablation study among differential-equation-based continuous models. _The results show that through generalization, our model achieves an advantage._

---

> > ### Comment · Reviewer_Qt1E · 2025-08-01
> > **Thank you**
> >
> > Thank you for your clarification. The responses have addressed most of my concerns. I will consider raising my score.

---

> > > ### Author Response · Authors · 2025-08-01
> > > **glad to hear that the responses have addressed most of the reviewer's concerns**
> > >
> > > Dear Reviewer Qt1E,
> > >
> > > Thank you for reconsidering our manuscript and for considering raising its rating. We are delighted that our responses have addressed your concerns. We sincerely appreciate the insightful feedback and the time you have dedicated to reviewing our work. Your comments—and our corresponding clarifications and revisions—have definitely enhanced the quality of the paper.
> > >
> > > Sincerely,
> > > The Authors of Paper 19675

---

### Official Review · Reviewer_ikK2 · 2025-07-17

**Clarity:** 4
**Significance:** 4
**Originality:** 4
**Rating:** 6
**Confidence:** 4

**Summary:**

NeuralODEs have shown success in integrating differential equations. However, the integer order neuralODEs do not implicitly by construction account for memory effects or historical dependencies. Motivated by Fractional calculus which is capable of accounting for memory effects and historical dependencies, the authors propose a generalised kernel-attention fractional Neural ODE framework (KatFDE). While fractional neural networks use statich memory weightings that are not learnable or adaptive, the authors replace the fixed / static kernels with neural attention kernels that adaptively weight historical dependencies better and lay better emphasis on important temporal contributors. Extensive evaluation has been carried out on graph learning problems and spatio-temporal traffic flow forecasting tasks and the KatFDE was shown to significantly outperform integer order neuralODEs and existing fractional approaches such as NvoFDE, FROND, etc.

**Questions:**

The only question I have is, have the authors considered testing KatFDE spatio-temporally continuous non-linear temporally evolving systems such as fluid flows / heat conduction/ black-scholes/reaction-diffusion/allen-cahn systems? Would have been a nice plethora of systems to evaluate the KatFDE framework on to know where KatFDE is most likely applicable or even establish if KatFDE is universally applicable with more concrete evidence. **Actionable:** Please provide a set of limitations, possible extensions in a separate paragraph in the conclusion.

**Ethical Concerns:**

["NO or VERY MINOR ethics concerns only"]

**Final Justification:**

My score remains unchanged but I have recommended a strong accept of this work as most of my concerns have been addressed in the rebuttal.
1. Additional clear test results were presented for different spatio-temporally/ temporally evolving non-linear systems which showed that KatFDE is a clear win amongst the other methods presented.
2. As recommended, the limitations of KatFDE in terms of being able to capture spatial interactions has also been addressed.

Importantly, KatFDE architecture would be a novel and valuable addition in the galore of methods to work with temporally evolving nonlinear systems.

**Limitations:**

The limitations if any, are not addressed by the authors towards the end of the paper. Would like the authors to highlight what other open questions remain to make KatFDE universally applicable for spatio-temporal systems (Discrete/ Continuous)

**Quality:**

4

**Strengths And Weaknesses:**

**Strengths:**
1. Strong mathematical background presented for the development of the KatFDE modeling framework. Convergence criterion, the kernel attention definitions, network construction. training methodology, and the overall modelling framework.
2. A thorough literature survey presented to identify the gaps and lacunae in the literature and the objectives have been well constructed around the gaps with sufficient motivation.
3. The numerical experiments are well detailed and a plethora of baseline models from literature are compared against technically sound evaluation metrics.
4. The capability of modelling temporal dependencies, and memory effects is extremely critical when modelling strongly non-linear, non-ideal and temporally evolving systems. KatFDE would be a useful framework in such scenarios.

---

> ### Author Rebuttal · Authors · 2025-07-31
>
> Thank you for your careful review of our paper. We are grateful for your recognition of our novelty.
>
> Furthermore, we would be honored if you could lend your insights during the discussion phase. Your perspective would be invaluable in clarifying any potential misunderstandings about our work and in contributing to a more robust discussion.
>
>
>
> ## Testing Other Spatio-temporally Continuous Non-linear Temporally Evolving Systems
>
> We thank the reviewer for this suggestion. To further demonstrate KatFDE’s generality and capability, we have added three additional experiments on complex temporally evolving systems.
>
> __1. Fluid Flow Prediction:__
>
> We evaluated KatFDE on turbulent boundary-layer flow [H1], with velocity fields measured using particle image velocimetry at five Reynolds numbers (Re = 600, 980, 1370, 1780, 2220), each containing approximately 6,000 snapshots. We trained on four Reynolds numbers and tested on the fifth. Models observed 6 consecutive snapshots to predict the next 6, using a CNN encoder–decoder for spatial feature extraction and LSTM, Transformer, Neural ODE, or our KatFDE for latent state temporal evolution.
> Turbulent flows exhibit strong memory effects due to the cascade of energy across different scales and the persistence of coherent structures. The complex temporal evolution of turbulence makes it an ideal test case for our attention-based fractional framework, which can adaptively weight historical flow states based on their relevance to current dynamics. The preliminary prediction results are summarized in **Table R1**, demonstrating that KatFDE's adaptive memory mechanism effectively captures the nonlinear complexity inherent in turbulent flows, outperforming other approaches.
>
> | Model       | LSTM+CNN | Transformer+CNN | ODE+CNN | Ours+CNN |
> |--------------|----------|-----------------|---------|----------|
> | RMSE     | 1.5035   | 0.4701           | 0.5962  | **0.3215** |
> | MAE      | 0.6027   | 0.3532           | 0.2799  | **0.2041** |
>
> **Table R1**: Average prediction error of different models on turbulent vector field prediction.
>
> [H1] Towne, A., Dawson, S., Brès, G. A., Lozano-Durán, A., Saxton-Fox, T., Parthasarthy, A., Biler, H., Jones, A. R., Yeh, C.-A., Patel, H., Taira, K. (2022). A database for reduced-complexity modeling of fluid flows. AIAA Journal 61(7): 2867-2892.
>
>
> __2. Urban Population Mobility Prediction:__
>
> Post-disaster urban mobility dynamics exhibit complex patterns driven by both the disruptive disaster context and underlying habitual mobility. We evaluated KatFDE on Florida’s population mobility data during Hurricane Dorian (August 1–September 10, 2019), sourced from SafeGraph and reported in [H2].
> This dataset records daily inter-regional movements within Florida. We compared KatFDE against baselines including LSTM, Transformer, and neural ODE variants. As shown in **Table R2**, KatFDE achieves state-of-the-art performance.
> This is a clear use case for KatFDE: the fractional operator enables long memory, while the adaptive attention kernel allows dynamic weighing of recent disaster-related vs. routine historical patterns. The model can down-weight pre-disaster commuting patterns during the hurricane and re-integrate them during recovery.
>
> | Model         | MAE         | NRMSE  | R²     |
> |--------------|-------------|--------|--------|
> | LSTM          | 302884.0938 | 0.6689 | 0.5526 |
> | AGCRN         | 273478.9062 | 0.5377 | 0.7109 |
> | NDCN          | 406064.2812 | 0.3987 | 0.8411 |
> | CG-ODE        | 224787.1250 | 0.2936 | 0.9138 |
> | STG-NCDE      | 226016.1562 | 0.6683 | 0.5533 |
> | PatchTST      |  96736.0547 | 0.1632 | 0.9734 |
> | CDGON         |  59767.4805 | 0.0724 | 0.9948 |
> | **KatFDE (ours)** | **24040.9355** | **0.0506** | **0.9974** |
>
> **Table R1**: Average prediction error of different models on post-disaster urban mobility.
>
> [H2] Li, J., etc. Physics-informed neural ode for post-disaster mobility recovery. In KDD 2024.
>
> __3. Biological Neural Spike Train Dynamics:__
>
> We next tested KatFDE on biological time-series data of neural spike trains from multiple animals and brain regions. Neural systems are inherently history-dependent: the evolution of a neuron's membrane potential is influenced by prior inputs and spike events, and downstream firing patterns are shaped by this accumulated temporal context.
> We tested the experimental results on spiking datasets Allen and Retina [H3]. The Allen dataset contains spike train data from various brain regions and is designed to evaluate models for temporal and spatiotemporal neural activity classification. The Retina dataset provides spike trains from salamander retinal ganglion cells under four visual stimuli for stimulus-type classification.  As **Table R3** shows, KatFDE outperforms standard recurrent and neural ODE models, highlighting its advantage in modeling nonlinear and temporally dependent neural systems.
>
>
> | Dataset | LSTM  | Neural ODE | KatFDE (ours)    |
> |--------|--------|------------------|---------|
> | Allen  | 85.05 | 85.05      | **86.03** |
> | Retina    | 90.36| 92.25    | **94.79** |
>
> **Table R3**: Test Accuracy of different models on neuron spike train classification (%)
>
>
> [H3] Lazarevich I, Prokin I, Gutkin B, et al. Spikebench: An open benchmark for spike train time-series classification[J]. PLOS Computational Biology, 2023, 19(1): e1010792.
>
>
> ### Limitations Discussion
>
> We will include the following Limitations Discussion in the paper conclusion part:
>
> There are some limitations of KatFDE.
> First, KatFDE is based on a deterministic equation. In practice, we often need to handle stochastic data and outputs with confidence intervals, which the current framework does not support.
> Second, the spatial interaction (i.e., the spatial partial differential components) at a given time is not explicitly included in the framework with direct incorporation of physical laws. In this paper, we rely on other modules, such as GNN and CNN, to handle spatial interactions. A more natural and deeper fusion of these components may be explored in future work.
> Third, KatFDE may require higher computational cost when using nonlocal kernels.
> To make KatFDE more generally applicable, future work should explicitly include spatial interactions and consider stochastic models instead of a purely deterministic formulation. In addition, designing more efficient numerical solvers and incorporating more efficient attention mechanisms will help reduce computation and memory usage, especially for large-scale datasets.

---

### Decision · Program_Chairs · 2025-09-17

**Decision:**

Accept (poster)

**Comment:**

This paper introduces KatFDE, a generalised kernel-attention fractional Neural ODE framework motivated by fractional calculus. The key idea is to replace fixed power-law memory kernels in traditional fractional differential equations with a flexible, learnable kernel based on attention mechanisms. This design unifies integer-order, variable-order, and Caputo dynamics within a single continuous-depth framework, supported by rigorous theoretical analysis. The authors establish boundedness under both singular and nonsingular cases and prove well-posedness of the neural integral equations using Banach fixed-point arguments, ensuring uniqueness of solutions. While the scaled dot-product attention mechanism itself is not novel, its use as a learnable kernel within fractional derivatives is a fresh and meaningful contribution.

The empirical evaluation is strong. The numerical experiments are comprehensive and compare KatFDE against a wide range of baselines using appropriate metrics. The results demonstrate KatFDE’s ability to effectively model long-term temporal dependencies and memory effects, which is particularly important in non-linear, non-ideal, and evolving systems.

The presentation is carefully crafted to bridge mathematics and machine learning audiences. The gradual introduction of fractional calculus concepts—from first-order to variable-order definitions—makes the paper accessible without sacrificing rigor. This pedagogical approach strengthens the work’s potential impact across communities.